# Statistical downscaling with the downscaleR package (v3.1.0): Contribution to the VALUE intercomparison experiment

Joaquín Bedia[1], Jorge Baño-Medina[2], Mikel N. Legasa[2], Maialen Iturbide[2], Rodrigo Manzanas[1], Sixto Herrera[1], Ana Casanueva[1], Daniel San-Martín[3], Antonio S. Cofiño[1], and José Manuel Gutiérrez[2]

[1]Meteorology Group. Dpto. de Matemática Aplicada y Ciencias de la Computación. Universidad de Cantabria, Santander, 39005, Spain
[2]Meteorology Group. Instituto de Física de Cantabria (CSIC - Universidad de Cantabria), Santander, 39005, Spain
[3]Predictia Intelligent Data Solutions, Santander, 39005, Spain

**Correspondence:** Joaquín Bedia (bediaj@unican.es)

**Abstract.** The increasing demand for high-resolution climate information has attracted a growing attention for statistical downscaling methods (SD), due in part to their relative advantages and merits as compared to dynamical approaches (based on regional climate model simulations), such as their much lower computational cost and their fitness-for-purpose for many local-scale applications. As a result, a plethora of SD methods is nowadays available for climate scientists, which has motivated recent efforts for their comprehensive evaluation, like the VALUE initiative (http://www.value-cost.eu). The systematic inter-comparison of a large number of SD techniques undertaken in VALUE, many of them independently developed by different authors and modeling centers in a variety of languages/environments, has shown a compelling need for new tools allowing for their application within an integrated framework. With this regard, downscaleR is an R package for statistical downscaling of climate information which covers the most popular approaches (Model Output Statistics —including the so called 'bias correction' methods— and Perfect Prognosis) and state-of-the-art techniques. It has been conceived to work primarily with daily data and can be used in the framework of both seasonal forecasting and climate change studies. Its full integration within the climate4R framework (Iturbide et al., 2019) makes possible the development of end-to-end downscaling applications, from data retrieval to model building, validation and prediction, bringing to climate scientists and practitioners a unique comprehensive framework for SD model development.

In this article the main features of downscaleR are showcased through the replication of some of the results obtained in VALUE, making an emphasis in the most technically complex stages of perfect-prog model calibration (predictor screening, cross-validation and model selection) that are accomplished through simple commands allowing for extremely flexible model tuning, tailored to the needs of users requiring an easy interface for different levels of experimental complexity. As part of the open-source climate4R framework, downscaleR is freely available and the necessary data and R scripts to fully replicate the experiments included in this paper are also provided as a companion notebook.

# 1 Introduction

Global Climate Models (GCMs) —atmospheric, coupled oceanic-atmospheric, and earth system models— are the primary tools used to generate weather and climate predictions at different forecast horizons, from intra-seasonal to centennial scales. However, raw model outputs are often not suitable for climate impact studies due to their limited resolution (typically hundreds of kilometers) and the presence of biases in the representation of regional climate (Christensen et al., 2008), attributed to a number of reasons such as the imperfect representation of physical processes and the coarse spatial resolution that does not permit an accurate representation of small-scale processes. To partially overcome these limitations, a wide variety of *downscaling* techniques have been developed, aimed at bridging the gap between the coarse-scale information provided by GCMs and the regional/local climate information required for climate impact and vulnerability analysis. To this aim both dynamical (based on regional climate models, RCMs; see, e.g. Laprise, 2008) and empirical/statistical approaches have been introduced during the last decades. In essence, statistical downscaling (SD, Maraun and Widmann, 2018) methods rely on the establishment of a statistical link between the local-scale meteorological series (predictand) and large-scale atmospheric variables at different pressure levels (predictors, e.g.: geopotential, temperature, humidity . . . ). The statistical models/algorithms used in this approach are first calibrated using historical (observed) data of both coarse predictors (reanalysis) and local predictands for a representative climatic period (usually a few decades) and then applied to new (e.g., future or retrospective) global predictors (GCM outputs) to obtain the corresponding locally downscaled predictands (von Storch et al., 1993). SD techniques were first applied in short-range weather forecast (Klein et al., 1959; Glahn and Lowry, 1972) and later adapted to larger prediction horizons, including seasonal forecasts and climate change projections, being the latter problem the one that has received the most extensive attention in the literature. SD techniques are often also applied to RCM outputs (usually referred to as 'hybrid downscaling', e.g., Turco and Gutiérrez, 2011), and therefore both approaches (dynamical and statistical) can be regarded as complementary rather than mutually exclusive .

Notable efforts have been done in order to assess the credibility of regional climate change scenarios. In the particular case of SD, a plethora of methods exists nowadays, and a thorough assessment of their intrinsic merits and limitations is required to guide practitioners and decision-makers with credible climate information (Barsugli et al., 2013). In response to this challenge, the COST Action VALUE (Maraun et al., 2015, http://www.value-cost.eu) is an open collaboration that has established a European network to develop and validate downscaling methods, fostering collaboration and knowledge exchange between dispersed research communities and groups, with the engagement of relevant stakeholders (Rössler et al., 2019). VALUE has undertaken a comprehensive validation and intercomparison of a wide range of SD methods (over 50), representative of the most common techniques covering the three main approaches, namely perfect prognosis, model output statistics — including bias correction— and weather generators (Gutiérrez et al., 2019). VALUE also provides a common experimental framework for statistical downscaling and has developed community-oriented validation tools specifically tailored for the systematic validation of different quality aspects that had so far received little attention (see Maraun et al., 2019b, for an overview), such as the ability of the downscaling predictions to reproduce the observed temporal variability (Maraun et al.,

2019a), the spatial variability among different locations (Widmann et al., 2019), reproducibility of extremes (Hertig et al., 2019) and process-based validation (Soares et al., 2019).

The increasing demand for high-resolution predictions/projections for climate impact studies, and the relatively fast development of SD in the last decades, with a growing number of algorithms and techniques available, has motivated the development of tools for bridging the gap between the inherent complexities of SD and the user's needs, able to provide end-to-end solutions in order to link the outputs of the GCMs and ensemble prediction systems to a range of impact applications. One pioneer service was the interactive, web-based Downscaling Portal (Gutiérrez et al., 2012) developed within the EU-funded ENSEMBLES project (van der Linden and Mitchell, 2009), integrating the necessary tools and providing the appropriate technology for distributed data access and computing, enabling user-friendly development and evaluation of complex SD experiments for a wide range of alternative methods (analogs, weather typing, regression ...). The downscaling portal is in turn internally driven by MeteoLab, (https://meteo.unican.es/trac/MLToolbox/wiki), an open-source Matlab™ toolbox for statistical analysis and data mining in meteorology, focused on statistical downscaling methods.

There are other existing tools available for the R computing environment implementing SD methods (beyond the most basic MOS and 'bias correction' techniques not addressed in this study, but see Sec. 2), like the R package `esd` (Benestad et al., 2015), freely available from the Norwegian Meteorological Institute (MET Norway). This package provides utilities for data retrieval and manipulation, statistical downscaling and visualization, implementing several classical methods (EOF analysis, regression, canonical correlation analysis, multi-variate regression and weather generators, among others). A more specific downscaling tool is provided by the package `Rglimclim` (https://www.ucl.ac.uk/~ucakarc/work/glimclim.html), a multivariate weather generator based on generalised linear models (see Sec. 2.2) focused on model fitting and simulation of multisite daily climate sequences, including the implementation of graphical procedures for examining fitted models and simulation performance (see e.g. Chandler and Wheater, 2002).

More recently, the `climate4R` framework (Iturbide et al., 2019), based on the popular R language (R Core Team, 2019) and other external open-source software components (NetCDF-Java, THREDDS etc.), has also contributed with a variety of methods and advanced tools for climate impact applications, including statistical downscaling. `climate4R` is formed by different seamlessly integrated packages for climate data access, processing (e.g. collocation, binding, and subsetting), analysis and visualization, tailored to the needs of the climate impact assessment communities in various sectors and applications, including comprehensive metadata and output traceability (Bedia et al., 2019), and provided with extensive documentation, wiki pages and worked examples (notebooks) allowing reproducibility of several research papers (see e.g.: https://github.com/SantanderMetGroup/notebooks). Furthermore, the *climate4R Hub* is a cloud-based computing facility that allows to run `climate4R` on the cloud using docker and jupyter-notebook (https://github.com/SantanderMetGroup/climate4R/tree/master/docker). The `climate4R` framework is presented by Iturbide et al. (2019), and some of its specific components for sectoral applications are illustrated e.g. in Cofiño et al. (2018) —seasonal forecasting—, Frías et al. (2018) —visualization—, Bedia et al. (2018) —forest fires—, or Iturbide et al. (2018) —species distributions— among others. In this context, the R package `downscaleR` has been conceived as a new component of `climate4R` to undertake SD exercises, allowing for a straightforward application of a wide range of methods. It builds on the previous experience of the MeteoLab

Toolbox in the design and implementation of advanced climate analysis tools, and incorporates novel methods and enhanced functionalities implementing the state-of-the-art SD techniques to be used in forthcoming intercomparison experiments in the framework of the EURO-CORDEX initiative (Jacob et al., 2014), in which the VALUE activities have merged and will follow on. As a result, unlike previous existing SD tools available in R, `downscaleR` is integrated within a larger climate processing framework providing end-to-end solutions for the climate impact community, including efficient access to a wide range of data formats, either remote or locally stored, extensive data manipulation and analysis capabilities, and export options to common geoscientific file formats (such as netCDF), thus providing maximum interoperability to accomplish successful SD exercises in different disciplines and applications.

This paper introduces the main features of `downscaleR` for *perfect-prognosis* statistical downscaling (as introduced in Sec. 2) using to this aim some of the methods contributing to VALUE. The particular aspects related to data preprocessing (predictor handling, etc.), SD model configuration, and downscaling from GCM predictors are described, thus covering the whole downscaling cycle from the user's perspective. In order to showcase the main `downscaleR` capabilities and its framing within the ecosystem of applications brought by `climate4R`, the paper reproduces some of the results of the VALUE intercomparison presented by Gutiérrez et al. (2019), using public datasets (described in Sec. 3.1), and considering two popular SD techniques (analogs and generalized linear models), described in Sec. 2.2. The `downscaleR` functions and the most relevant parameters used in each experiment are shown in Sections 3.3 and 4, after a schematic overview of the different stages involved in a typical *perfect-prog* SD experiment (Sec. 2.1). Finally in Sec. 4.2, locally downscaled projections of precipitation for a high emission scenario (RCP 8.5) are calculated for the future period 2071-2100 using the output from one state-of-the-art GCM contributing to the CMIP5 Project.

## 2 Perfect-prognosis Statistical Downscaling (SD): **downscaleR**

The application of SD techniques to the global outputs of a GCM (or RCM) typically entails two phases. In the training phase, the model parameters (or algorithms) are fitted to data (or tuned/calibrated) and cross-validated using a representative historical period (typically a few decades) with existing predictor and predictand data. In the downscaling phase, which is common to all SD methods, the predictors given by the GCM outputs are plugged into the models (or algorithms) to obtain the corresponding locally downscaled values for the predictands. According to the approach followed in the training phase, the different SD techniques can be broadly classified into two categories (Rummukainen, 1997; Marzban et al., 2006, also see Maraun and Widmann (2018) for a discussion on these approaches), namely Perfect Prognosis (PP) and Model Output Statistics (MOS). In the PP approach, the statistical model is calibrated using observational data for both the predictands and predictors (see, e.g., Charles et al., 1999; Timbal et al., 2003; Bürger and Chen, 2005; Haylock et al., 2006; Fowler et al., 2007; Hertig and Jacobeit, 2008; Sauter and Venema, 2011; Gutiérrez et al., 2013). In this case, 'observational' data for the predictors is taken from a reanalysis (which assimilates day by day the available observations into the model space). In general, reanalyses are more constrained by assimilated observations than by internal model variability and thus can reasonably assumed to reflect 'reality' (Sterl, 2004). The term 'perfect' in PP refers to the assumption that the predictors are bias-free. This assumption is generally

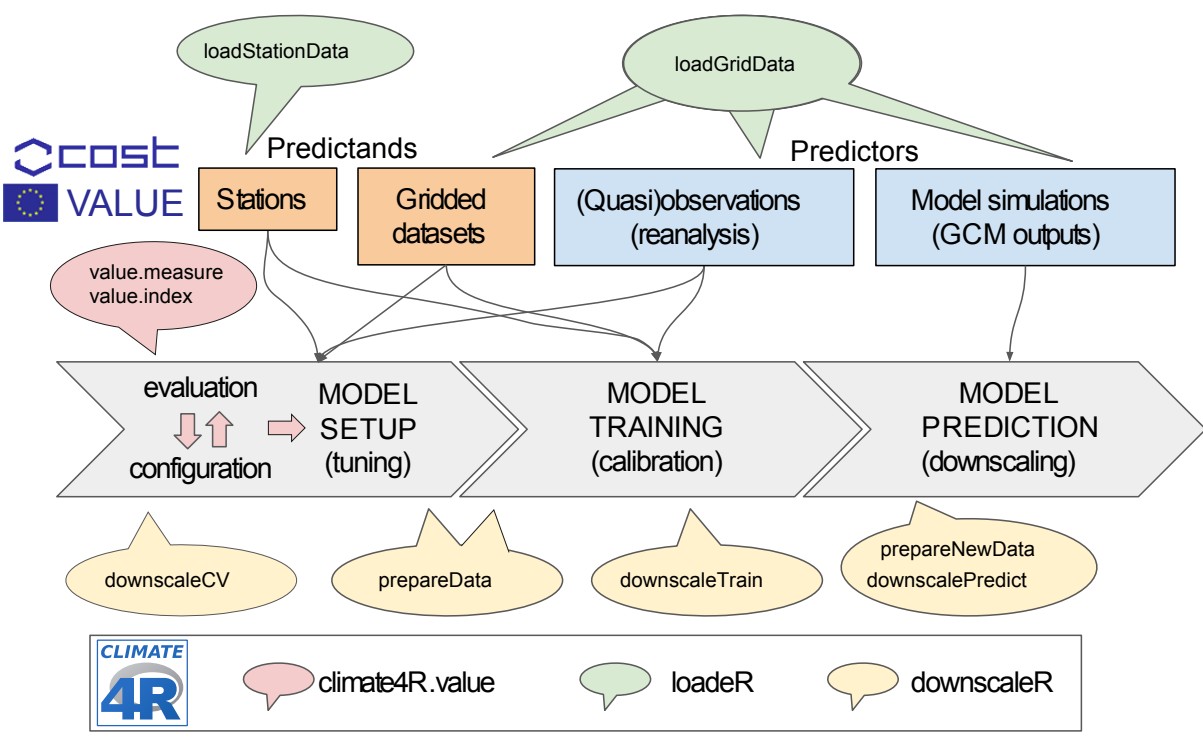

**Figure 1.** Schematic overview of the R package `downscaleR` and its framing into the `climate4R` framework for climate data access and analysis. The typical perfect-prog downscaling phases are indicated by the grey arrows. i) In first place, model setup is undertaken. This process is iterative and usually requires testing many different model configurations under a cross-validation set up until an optimal configuration is achieved. The `downscaleCV` function (and `prepareData` under the hood) is used in this stage for a fine-tuning of the model. Model selection is determined through the use of indices and measures reflecting model suitability for different aspects that usually depend on specific research aims (e.g. good reproducibility of extreme events, temporal variability, spatial dependency across different locations ...). The validation is achieved through the `climate4R.value` package (red-shaded callout), implementing the VALUE validation framework. ii) Model training: once an optimal model is achieved, model training is performed using the `downscaleTrain` function. iii) Finally, the calibrated model is used to undertake downscaling (i.e. model predictions) using the function `downscalePredict`. The data to be used in the predictions requires appropriate pre-processing (e.g. centering and scaling using the predictor set as reference, projection of PC's onto predictor EOF's, etc.) that is performed under the hood by function `prepareNewData` prior to model prediction with `downscalePredict`.

accepted (albeit it may hold not true in the tropics, see e.g. Brands et al. (2012)). As a result, in the PP approach predictors and predictand preserve day-to-day correspondence. Unlike PP, in the MOS approach the predictors are taken from the same GCM (or RCM) for both the training and downscaling phases. For instance, in MOS approaches, local precipitation is typically downscaled from the direct model precipitation simulations (Widmann et al., 2003). In weather forecasting applications MOS techniques also preserve the day-to-day correspondence between predictors and predictand but, unlike PP, this doesn't hold true

in a climate context. As a result, MOS methods typically work with the (locally-interpolated) predictions and observations of the variable of interest (a single predictor). In MOS, the limitation of having homogeneous predictor-predictand relationships applies only in a climate context, and therefore many popular 'bias correction' techniques (e.g. linear scaling, quantile-quantile mapping etc.) lie in this category. In this case, the focus is on the statistical similarity between predictor and predictand, and there is no day-to-day correspondence of both series during the calibration phase. The application of MOS techniques in a climate context using `downscaleR` is already shown in Iturbide et al. (2019). Here, the focus is done on the implementation of PP methods, that entail greater technical complexities for their application from a user's perspective, but have received less attention from the side of climate service development. A schematic diagram showing the main phases of perfect-prog downscaling is shown in Fig. 1.

## 2.1 SD model setup: Configuration of predictors

As general recommendations, a number of aspects need to be carefully addressed when looking for suitable predictors in the PP approach (Wilby et al., 2004; Hanssen-Bauer et al., 2005): i) The predictors should account for a major part of the variability in the predictands, ii) the links between predictors and predictands should be temporally stable/stationary and iii) the large-scale predictors must be realistically reproduced by the global climate model. Since different global models are used in the calibration and downscaling phases, large-scale circulation variables well represented by the global models are typically chosen as predictors in the PP approach, whereas variables directly influenced by model parametrizations and/or orography (e.g. precipitation) are usually not considered. For instance, predictors generally fulfilling these conditions for downscaling precipitation are humidity, geopotential or air temperature (see Sec. 3.1.2) at different surface pressure vertical levels. Only sea-level pressure and 2 m air temperature are usually used as near-surface surface predictors. An example of evaluation of this hypothesis is later presented in Sec. 4.2.1 of this study. Often, predictors are proxies for physical processes, which is a main reason for non-stationarities in the predictor/predictand relationship, as amply discussed in Maraun and Widmann (2018). Furthermore, reanalysis choice has been reported as an additional source of uncertainty for SD model development (Brands et al., 2012), although its effect is of relevance only in the tropics (see e.g.: Manzanas et al., 2015). In regard with the assumption ii.), predictor selection and the training of transfer functions are carried out on short term variability in present climate, whereas the aim is typically to simulate long term changes of short term variability (Huth, 2004; Maraun and Widmann, 2018), which limits the performance of PP and makes it particularly sensitive to the method type and the predictor choice (Maraun et al., 2019b).

For all these reasons, the selection of informative and robust predictors during the calibration stage is a crucial step in SD modelling (Fig. 1), being model predictions very sensitive to the strategy used for predictor configuration (see e.g. Benestad, 2007; Gutiérrez et al., 2013). PP techniques can consider point-wise and/or spatial-wise predictors, using either the raw values of a variable over a region of a user-defined extent or only at nearby grid boxes and/or the Principal Components (PCs) corresponding to the Empirical Orthogonal Functions (EOFs, Preisendorfer, 1988) of the variables considered over a representative geographical domain (which must be also conveniently determined). Usually, the latter are more informative in those cases where the local climate is mostly determined by synoptic phenomena whereas the former may be needed to add some informa-

tion about the local variability in those cases where small-scale processes are important (see e.g. Benestad, 2001). Sometimes, both types of predictors are combined in order to account for both synoptic and local effects. In this sense, three non-mutually exclusive options are typically used in downscaling experiments next summarized:

1. Using raw atmospheric fields for a given spatial domain, typically continental- or national-wide for downscaling monthly and daily data, respectively. For instance, in the VALUE experiment, predefined subregions within Europe are used for training (Fig. 2), thus helping to reduce the dimension of the predictor set. Alternatively, step-wise or regularized methods can be used to automatically select the predictor set from the full spatial domain.

   2. Using principal components obtained from these fields (Benestad, 2001). Working with PCs allows to filter-out high
frequency variability which may be not properly linked to the local-scale, greatly reducing the dimensionality of the problem related to the deletion of redundant and/or colinear information from the raw predictors. These predictors convey large-scale information to the predictor set, and are often also referred to as 'spatial predictors'. These can be either a number of principal components calculated upon each particular variable (e.g. explaining 95% of the variability), and/or a combined PC calculated upon the (joined) standardized predictor fields ('combined' PCs).

3. The spatial extent of each predictor field may have a strong effect on the resulting model. Some variables of the predictor set may have explanatory power only nearby the predictand locations, while the useful information is diluted when considering larger spatial domains. As a result, it is common practice to include local information in the predictor set by considering only a few gridpoints around the predictand location for some of the predictor variables (this can be just the closest grid point or a window of a user-defined width). This category can be regarded as a particular case of point 1,
but considering a much narrower window centered around the predictand location. This local information is combined with the 'global' information provided by other global predictors (either raw fields —case 1— or principal components —case 2—) encompassing a larger spatial domain.

Therefore, predictor screening (i.e. variable selection) and their configuration is one of the most time-consuming tasks in perfect-prog experiments due to the potentially huge number of options required for a fine-tuning of the predictor set (spatial,
local or a combination of both, number of principal components and methodology for their generation etc.). As a result, SD model tuning is iterative and usually requires testing many different model configurations until an optimal one is attained (see e.g. Gutiérrez et al., 2013), as next described in Sec. 2.3. This requires a flexible, yet easily configurable interface, enabling users to launch complex experiments for testing different predictor setups in a straightforward manner. In `downscaleR`, the function `prepareData` has been designed to this aim, providing maximum user flexibility for the definition of all types of
predictor configurations with a single command call, building upon the raw predictor information (see Sec. 3.3).

## 2.2   Description of SD methods

`downscaleR` implements several PP techniques, ranging from the classical analogs and regression to more recent and sophisticated machine learning methods (Baño-Medina et al., 2019). For brevity, in this study we focus on the standard approaches

contributing to the VALUE intercomparison, namely analogs, linear models and generalized linear models, next briefly intro-
duced; the up-to-date description of methods is available at the `downscaleR` wiki (https://github.com/SantanderMetGroup/
downscaleR/wiki). All the SD methods implemented in `downscaleR` are applied using unique workhorse functions such as
`downscaleCV` (cross-validation), `downscaleTrain` (for model training), `downscalePredict` (for model prediction),
etc. (Fig. 1), that receive the different tuning parameters for each method chosen, providing maximum user flexibility for the
definition and calibration of the methods. Their application will be illustrated throughout Sections 3.3 and 4.

### 2.2.1 Analogs

This is a non-parametric analog technique (Lorenz, 1969; Zorita and von Storch, 1999), based on the assumption that similar
(or *analog*) atmospheric patterns (predictors) over a given region lead to similar local meteorological outcomes (predictand).
For a given atmospheric pattern, the corresponding local prediction is estimated according to a determined similarity measure
(tipically the Euclidean norm, which has been shown to perform satisfactorily in most cases, see e.g.: Matulla et al., 2008) from
a set of analog patterns within a historical catalog over a representative climatological period. In PP, this catalog is formed by
reanalysis data. In spite of its simplicity, analog performance is competitive against other more sophisticated techniques (Zorita
and von Storch, 1999), being able to take into account the non-linearity of the relationships between predictors and predictands.
Additionally, it is spatially coherent by construction, preserving the spatial covariance structure of the local predictands as long
as the same sequence of analogs for different locations is used, being spatial coherence underestimated otherwise (Widmann
et al., 2019). Hence, analog-based methods have been applied in several studies both in the context of climate change (see,
e.g., Gutiérrez et al., 2013) and seasonal forecasting (Manzanas et al., 2017). The main drawback of the analog technique is
that it cannot predict values outside the observed range, being therefore particularly sensitive to the non-stationarities arising
in climate change conditions (Benestad, 2010), thus preventing from its application to the far future, when temperature and
directly related variables are considered (see e.g. Bedia et al., 2013).

### 2.2.2 Linear Models (LMs)

(Multiple) linear regression is the most popular downscaling technique for suitable variables (e.g., temperature), although
it has been also applied to other variables after suitable transformation (e.g., to precipitation, typically taking the cubic root).
Several implementations have been proposed including both spatial (PC) and/or local predictors. Moreover, automatic predictor
selection approaches (e.g., stepwise) have been also applied (see Gutiérrez et al., 2019, for a review).

### 2.2.3 Generalized Linear Models (GLMs)

They were formulated by Nelder and Wedderburn (1972) in the 1970's and are an extension of the classical linear regression
which allows to model the expected value of a random predictand variable whose distribution belongs to the exponential family
($Y$) through an arbitrary mathematical function called *link function* ($g$) and a set of unknown parameters ($\beta$), according to

$$E(Y) = \mu = g^{-1}(X\beta), \tag{1}$$

where $X$ is the predictor and $E(Y)$ the expected value of the predictand. The unknown parameters, $\beta$, can be estimated by maximum likelihood, considering a least-squares iterative algorithm.

GLMs have been extensively used for SD in climate change applications (e.g., Brandsma and Buishand, 1997; Chandler and Wheater, 2002; Abaurrea and Asín, 2005; Fealy and Sweeney, 2007; Hertig et al., 2013), and more recently, also used for seasonal forecasts (Manzanas et al., 2017). For the case of precipitation, a two-stage implementation (see, e.g., Chandler and Wheater, 2002) must be used given its dual (*occurrence/amount*) character. In this implementation, a GLM with Bernoulli error distribution and *logit* canonical link function (also known as *logistic regression*) is used to downscale precipitation occurrence ($0 =$ no rain, $1 =$ rain) and a GLM with gamma error distribution and *log* canonical link-function is used to downscale precipitation amount, considering wet days only. After model calibration, new daily predictions are given by simulating from a gamma distribution, whose shape parameter is fitted using the observed wet days in the calibration period.

Beyond the classical GLM configurations, `downscaleR` allows for using both deterministic and stochastic versions of GLMs. In the former, the predictions are obtained from the expected values estimated by both the GLM for occurrence (*GLMo*) and the GLM for amount (*GLMa*). In the *GLMo*, the continuous expected values $\in [0, 1]$ are transformed into binary ones as 1 (0) either by fixing a cutoff probability value (e.g., 0.5) or by choosing a threshold based on the observed predictand climatology for the calibration period (the latter is the default behaviour in `downscaleR`). On the contrary, for *GLMa*, the expected values are directly interpreted as rain amounts. Moreover, `downscaleR` gives the option of generating stochastic predictions for both the *GLMo* the and *GLMa*, which could be seen as a dynamic predictor-driven version of the inflation of variance used in some regression-based methods (Huth, 1999).

### 2.3 SD model validation

When assessing the performance of any SD technique it is crucial to properly cross-validate the results in order to avoid misleading conclusions about model performance due to artificial skill. This is typically achieved considering a historical period for which observations exist to validate against. $k$-fold and leave-one-out cross-validation are among the most widely applied validation procedures in SD experiments. In a $k$-fold cross-validation framework (Stone, 1974; Markatou et al., 2005), the original sample (historical period) is partitioned into $k$ equal-sized and mutually exclusive subsamples (folds). In each of the $k$ iterations, one of these folds is retained for test (prediction phase) and the remaining $k-1$ folds are used for training (calibration phase). The resulting $k$ independent samples are then merged to produce a single time-series covering the whole calibration period, which is subsequently validated against observations. When $k = n$ (being $n$ the number of observations), the $k$-fold cross-validation is exactly the leave-one-out cross-validation (Lachenbruch and Mickey, 1968). Another common approach is the simpler "holdout" method, that partitions the data into just two mutually exclusive subsets ($k = 2$), called the training and test (or holdout) sets. In this case, it is common to designate $2/3$ of the data as the training set and the remaining $1/3$ as the test set (see e.g. Kohavi, 1995).

Therefore, PP models are first cross-validated under 'perfect conditions' (i.e.: using reanalysis predictors) in order to evaluate their performance against real historical climate records, before being applied to 'non-perfect' GCM predictors. Therefore, the aim of cross-validation in the PP approach is to properly estimate, given a known predictor dataset (large-scale variables from

reanalysis), the performance of the particular technique considered, having an "upper-bound" for its generalization capability when applied to new predictor data (large-scale variables from GCM). The workhorse for cross-validation in `downscaleR` is the function `downscaleCV`, that adequately handles data partition to create the training and test data subsets according to the parameters specified by the user, being tailored to the special needs of statistical downscaling experiments (i.e. random temporal/spatial folds, leave-one-year-out, arbitrary selection of years as folds, etc.).

During the cross-validation process, one or several user-defined measures are used in order to assess model performance (i.e., to evaluate how "well" do model predictions match the observations), such as accuracy measures, distributional similarity scores, inter-annual variability, trend matching scores etc. In this sense, model quality evaluation is a multi-faceted task with many possible and often unrelated aspects to look into. Thus, validation ultimately consists of deriving specific *climate indices* from model output, comparing these indices to reference indices calculated from observational data and quantifying the mismatch with the help of suitable performance *measures* (Maraun et al., 2015). In VALUE, the term "index" is used in a general way, including not only single numbers (e.g. the $90^{th}$ percentile of precipitation, lag-1 autocorrelation etc.) but also vectors such as time series (for instance, a binary time series of rain/no rain). Specific "measures" are then computed upon the predicted and observed indices, for instance the difference (bias, predicted - observed) of numeric indices, or the correlation of time series (Sec. 3.3.9). A comprehensive list of indices and measures has been elaborated by the VALUE cross-cutting group in order to undertake a systematic evaluation of downscaling methods. The complete list is presented in the VALUE Validation Portal[1]. Furthermore, all the VALUE indices and measures have been implemented in R and collected in the package `VALUE` (https://github.com/SantanderMetGroup/VALUE), allowing for further collaboration and extension with other initiatives, as well as for research reproducibility. The validation tools available in VALUE have been adapted to the specific data structures of the `climate4R` framework (see Sec. 1) through the wrapping package `climate4R.value` (https://github.com/SantanderMetGroup/climate4R.value), enabling a direct application of the comprehensive VALUE validation framework to downscaling exercises with `downscaleR` (Fig. 1). A summary of the subset of VALUE indices and measures used in this study is presented in Table. 1.

## 3 Illustrative Case Study: The VALUE experiment

The VALUE initiative (Maraun et al., 2015) produced the largest-to-date intercomparison of statistical downscaling methods with over 50 contributing techniques. The contribution of `MeteoLab` (and `downscaleR`) to this experiment included a number of methods which are fully reproducible with `downscaleR`, as we show in this example. This pan-European contribution was based on previous experience over the Iberian domain (Gutiérrez et al., 2013; San-Martín et al., 2016), testing a number of predictor combinations and method's configurations. In order to illustrate the application of `downscaleR`, in this example we first revisit the experiment over Iberian domain (but considering the VALUE framework and data), showing the code undertaking the different steps (Sec. 3.3). Afterwards, the subset of methods contributing to VALUE is applied at a pan-European scale, including also results of future climate scenarios (Sec. 4).

---

[1]http://www.value-cost.eu/validationportal/app/#!indices

| Code | Description | Type |
|------|-------------|------|
| R01 | Relative frequency of wet days (precip $\geq$ 1mm) | index |
| Mean | Mean | index |
| SDII | Simple Daily Intensity Index | index |
| Skewness | Skewness | index |
| WWProb | Wet-wet transition probability (wet $\geq$1mm) | index |
| DWProb | Dry-wet transition probability (wet $\geq$1mm) | index |
| WetAnnualMaxSpell | Median of the annual wet ($\geq$1mm) spell maxima | index |
| DryAnnualMaxSpell | Median of the annual dry ($<$1mm) spell maxima | index |
| AnnualCycleAmp | Amplitude of the daily annual cycle | index |
| Var | Quasi-variance | index |
| ratio | Ratio predicted/observed | measure[1] |
| ts.rs | Spearman correlation | measure[2] |
| ts.RMSE | Root Mean Square Error | measure[2] |
| ts.ks | Two-sample Kolmogorov-Smirnov (KS) test statistic | measure[2,3] |
| ts.ks.pval | (corrected) P-value of the two sample KS test statistic | measure[2,3] |

**Table 1.** Summary of the subset of VALUE validation indices and measures used in this study. Their codes are consistent with the VALUE reference list (`http://www.value-cost.eu/validationportal/app/#!indices`), except for 'ts.ks.pval', that has been included later in the VALUE set of measures. The superindices in the measures indicate the input used to compute them: 1: a single scalar value, corresponding to the predicted and observed indices; 2: The original predicted and observed precipitation time series; 3: Transformed time series (centered anomalies or standardized anomalies).

In order to reproduce the results of the VALUE intercomparison, the VALUE datasets are used in this study (Sec. 3.1). In addition, future projections from a CMIP5 GCM are also used to illustrate the application of the downscaling methods to climate change studies. For transparency and full reproducibility, the datasets are public and freely available for download using the `climate4R` tools, as indicated in Sec. 3.2. Next, the datasets are briefly presented. Further information on the VALUE data characteristics is given in Maraun et al. (2015) and Gutiérrez et al. (2019), and also at their official download URL (http://www.value-cost.eu/data). The reference period considered for model evaluation in perfect conditions is 1979–2008. In the analysis of the GCM predictors (Sec. 4.2.1), this period is adjusted to 1979-2005 constrained by period of the historical experiment of the CMIP5 models (Sec. 3.1.3). The future period for presenting the climate change signal analysis is 2071-2100.

## 3.1 Datasets

### 3.1.1 Predictand data (weather station records)

The European station dataset used in VALUE has been carefully prepared in order to be representative of the different European climates and regions and with a reasonably homogeneous spatial density (Fig. 2). To keep the exercise as open as possible, the downloadable (blended) ECA&D stations (Klein Tank et al., 2002) was used. From this, a final subset of 86 stations

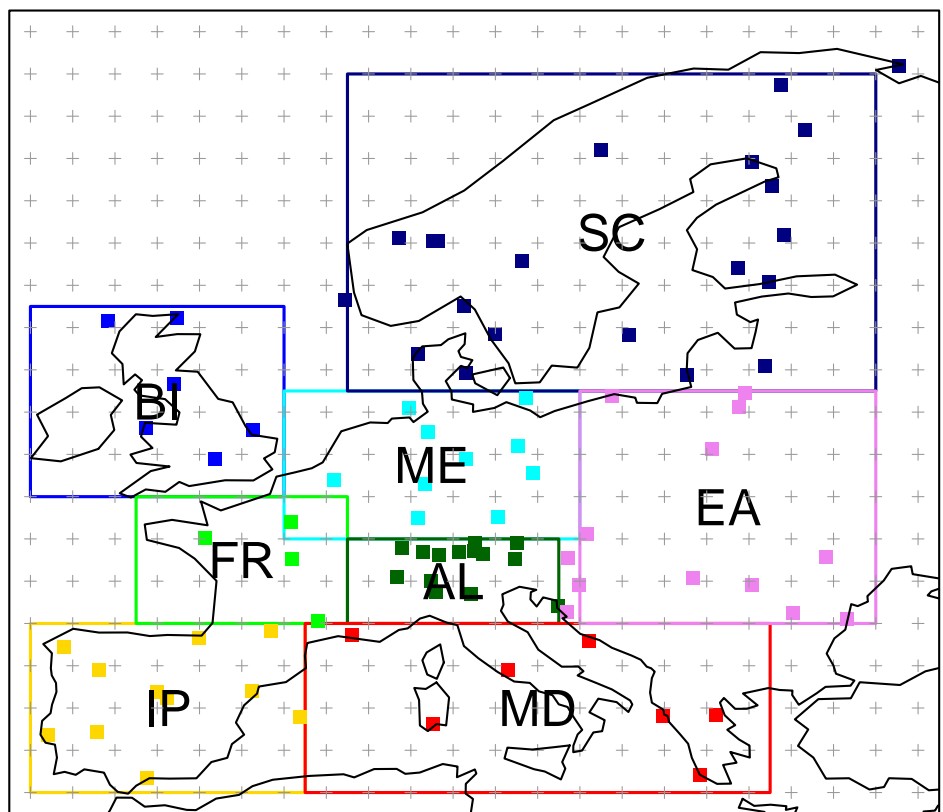

**Figure 2.** Location of the 86 stations of the ECA-VALUE-86 dataset (red squares). The coloured boxes show the eight PRU-DENCE subregions considered in the VALUE downscaling experiment for model training (Sec. 3.1). The regular grid of the predictor dataset, a $2 \times 2$ degrees resolution version of the ERA-Interim reanalysis, is also shown. The subregions considered are: IP (Iberian Peninsula), FR (France), BI (British Isles), MD (Mediterranean), AL (Alps), ME (Central Europe), SC (Scandinavia) and EA (Eastern Europe). Station metadata can be interactively queried through the VALUE Validation Portal application (`http://www.value-cost.eu/validationportal/app/#!datasets`).

was selected with the help of local experts in the different countries, restricted to high-quality stations with no more than 5% of missing values in the analysis period (1979-2008). Further details on predictand data preprocessing are provided in http://www.value-cost.eu/WG2_dailystations. The full list of stations is provided in Table 1 in Gutiérrez et al. (2019).

### 3.1.2  Predictor data (reanalysis)

In line with the experimental protocol of the Coordinated Regional Climate Downscaling Experiment (CORDEX, Giorgi et al., 310  2009), VALUE has used ERA-Interim (Dee et al., 2011) as the reference reanalysis to drive the experiment with 'perfect' predictors. For full comparability, the list of predictors used in VALUE is replicated in this study —see Table 2 in Gutiérrez

et al. (2019)—, namely: sea-level pressure, 2 meter air temperature, air temperature and relative humidity at 500,700 and 850 hPa surface pressure levels, and the geopotential height at 500 hPa.

The set of raw predictors corresponds to the full European domain shown in Fig. 2. The eight reference regions defined in the
315 PRUDENCE Project of model evaluation (Christensen et al., 2007) were used in VALUE as appropriate regional domains for training the models of the corresponding stations (Sec. 2.1). The stations falling within each domain are colored accordingly in Fig. 2.

### 3.1.3 Predictor data (GCM future projections)

In order to illustrate the application of SD methods to downscale future global projections from GCM predictors, here we
consider the outputs from the EC-EARTH model (in particular the *r12i1p1* ensemble member; EC-Earth Consortium, 2014), for the 2071-2100 period under the RCP8.5 scenario (Moss et al., 2010). This simulation is part of the CMIP5 intercomparison project (Taylor et al., 2011) and is officially served by the Earth System Grid Federation infrastructure (ESGF, Cinquini et al., 2014). In this study, data is retrieved from the Santander User Data Gateway (Sec. 4.2), which is the data access layer of the `climate4R` framework (described in Sec. 3.2).

| Method | ID | Predictor configuration description |
|--------|------|-------------------------------------|
| GLM | M1* | Spatial: $n$ combined PCs explaining 95% of variance |
| | M1-L | Spatial+local: $n$ combined PCs explaining 95% of variance + first nearest gridbox |
| | M2 | Spatial: $n$ independent PCs explaining 95% of the variance |
| | M3 | Local: first nearest gridbox |
| | M4 | Local: 4 nearest gridboxes |
| Analogs | M5 | Spatial: original standardized[†] predictor fields |
| | M6* | Spatial: $n$ combined PCs explaining 95% of variance |
| | M6-L | Local: 25 nearest gridboxes |
| | M7 | Spatial: $n$ independent PCs explaining 95% of the variance |

**Table 2.** Summary of predictor configurations tested. Local predictors always correspond to the original predictor fields previously standardized. Independent PCs are calculated separately for each predictor field, while combined PCs are computed upon the previously joined predictor fields (see Sec. 2.1 for more details). [†]The standardization in M5 is performed by subtracting to each grid cell the overall field mean, so the spatial structure of the predictor is preserved. Methods marked with an asterisk (*) are included in the VALUE intercomparison, with the slight difference that in VALUE, a fixed number of 15 PCs is used, and here the number varies slightly until achieving the percentage of explained variance indicated (in any case, the differences are negligible in terms of model performance). Methods followed by the -L suffix (standing for 'Local') are used only in the pan-European experiment described in Sec. 4.

## 3.2  Data retrieval with `climate4R`

All the data required are (remotely) available under the `climate4R` framework. Reanalysis (Sec. 3.1.2) and GCM data (Sec. 3.1.3) are retrieved in this example from the User Data Gateway (UDG), the remote data access layer of `climate4R`. The UDG is a climate service providing harmonized remote access to a variety of popular climate databases exposed via a THREDDS OPeNDAP service (Unidata, 2006) and a fine-grained authorization layer (the THREDDS Administration Panel, TAP) developed and managed by the Santander Meteorology Group (http://www.meteo.unican.es/udg-tap). The package `loadeR` allows easy access to the UDG datasets directly from R. For brevity, the details regarding data retrieval are omitted here, being already described in the previous works by Cofiño et al. (2018) and Iturbide et al. (2019). Suffice it here to show how the login into the UDG (via TAP) is done at the beginning of the R session and how the different collocation parameters for data retrieval (including the dataset Id and the names of the variables and their vertical surface pressure levels) are passed to the function `loadGridData`. It is also useful to remind that the user has access to a full list of public datasets available through the UDG and their Id's using the helper function `UDG.datasets`, and that an inventory of all available variables for each dataset can be obtained using the function `dataInventory`.

First of all, the required `climate4R` packages are loaded, including package `transformeR`, that undertakes multiple generic operations of data manipulation and `visualizeR` (Frías et al., 2018), used for plotting. Specific instructions for package installation are provided in the Supplementary Notebook of this paper, and in the principal page of the `climate4R` repo at GitHub (https://github.com/SantanderMetGroup/climate4R). The code used in each section is interwoven with the text in `verbatim` fonts. Lengthy lines of code are continued in the following line after indentation.

```
library(loadeR)
library(transformeR)
library(visualizeR)
library(downscaleR)
library(climate4R.value)
```

### 3.2.1  Loading Predictor Data

```
loginUDG(username = "****", password = "****")
# Register at http://www.meteo.unican.es/udg-tap

vars <- c("psl","tas","ta@500","ta@700", "ta@850",
          "hus@500","hus@850","z@500")

# The bounding box of the Iberia region (IP) is extracted:

data("PRUDENCEregions", package = "visualizeR")
bb <- PRUDENCEregions["IP"]@bbox
lon <- bb[1,]; lat <- bb[2,]
```

```
grid.list <- lapply(variables, function(x) {
    loadGridData(dataset = "ECMWF_ERA-Interim-ESD",
                 var = x,
                 lonLim = lon,
                 latLim = lat,
                 years = 1979:2008)
    }
)
```

In `climate4R`, climate variables are stored in the so called data *grids*, following the *Grid Feature Type* nomenclature of the Unidata Common Data Model[2], on which the `climate4R` data access layer and its data structures are based on. In order to efficiently handle multiple variables used as predictors in downscaling experiments, 'stacks' of grids encompassing the same spatial (and by default also temporal) domain are used. These are known as *multiGrids* in `downscaleR`, and can be obtained using the constructor `makeMultiGrid` from a set of -dimensionally consistent- grids. Next, a multigrid is constructed with the full set of predictors:

```
x <- makeMultiGrid(grid.list)
```

### 3.2.2 Loading Predictand Data

The `VALUE` package, already presented in Sec. 2.3, gathers all the validation routines used in VALUE. For convenience, the station dataset ECA-VALUE-86 (described in Sec. 3.1.1) is a built-in. As package `VALUE` is a dependency of the wrapper package `climate4R.VALUE` (see Sec. 2.3), its availability as installed package is assumed here:

```
v86 <- file.path(find.package("VALUE"), "example_datasets",
                 "VALUE_ECA_86_v2.zip")
```

Stations are loaded with the function `loadStationData` from package `loadeR`, tailored to the standard ASCII format defined in `climate4R`, also adopted by the VALUE framework.

```
y <- loadStationData(dataset = v86, var = "precip",
                     lonLim = lon, latLim = lat,
                     years = 1979:2008)
```

Since the variable precipitation requires two-stage modelling using GLMs (occurrence —binary— and amount —continuous—, see Sec. 2.2), the original precipitation records loaded require transformation. The function `binaryGrid` undertakes this frequent operation. Also, all the values below 1 mm converted to zero (note the use of argument `partial` that sets to zero only the values not fulfilling the condition `"GE"`, that is, 'Greater or Equal' than the threshold value given).

```
y <- binaryGrid(y, condition = "GE", threshold = 1,
                partial = TRUE)
y_bin <- binaryGrid(y, condition = "GE", threshold = 1)
```

---

[2]https://www.unidata.ucar.edu/software/thredds/current/netcdf-java/tutorial/GridDatatype.html

Both raw predictors and predictand set are now ready for SD model development.

## 3.3    Worked-out Example for the Iberian Domain

Building on the previous work by San-Martín et al. (2016) regarding predictor selection for precipitation downscaling, a number of predictor configuration alternatives is tested here. For brevity, the experiment is restricted to one of the VALUE subregions (Iberia, Fig. 2), avoiding a recursive repetition of the code for the 8 domains (the full code is provided in the companion paper

notebook, see the Code and Data availability Section at the end of the manuscript). From the range of methods tested in San-Martín et al. (2016), the methods labeled as M1 and M6 in Table 2 were also used in the VALUE intercomparison (for every subregion) in order to use spatial predictors for GLM and Analog methods (these are labelled as GLM-DET and ANALOG in Table 3 of Gutiérrez et al. (2019) respectively). In the particular case of method M6, this is implemented in order to minimize the number of predictors by compressing the information with PCs, hence improving the computational performance of the

method by accelerating the analog search. The full list of predictor variables and the same reference period (1979–2008) used in VALUE (enumerated in Sec. 3.1.2) is here applied for all the configurations tested, that are summarized in Table 2 following the indications given in Sec. 2.1.

### 3.3.1    Method configuration experiment over Iberia

In this section, the different configurations of the above described techniques (Table 2) are used to produce local predictions

of precipitation. The experimental workflow is presented following the schematic representation of Fig. 1, so the different subsections roughly correspond to the main blocks therein depicted (the future downscaled projections from a GCM will be later illustrated in Sec. 4.2). We partially replicate here the results obtained by Gutiérrez et al. (2019), which are the methods labelled as M1 and M6.

     As indicated in Sec. 2.1, `prepareData` is the workhorse for predictor configuration. The function handles all the com-

plexities of the predictor configuration under the hood, receiving a large number of arguments affecting the different aspects of predictor configuration, that are internally passed to other `climate4R` functions performing the different tasks required (i.e. data standardization, principal component analysis, data subsetting etc.). Furthermore, `downscaleR` allows for a flexible definition of local predictors of arbitrary window width (including just the closest grid-point). As the optimal predictor configuration is chosen after cross-validation, typically the function `downscaleCV` is used in first place. The latter function makes

internal calls to `prepareData` recursively for the different training subsets defined.

     As a result, `downscaleCV` receives as input all the arguments of `prepareData` for predictor configuration as a list, plus other specific arguments controlling the cross-validation setup. For instance, the argument `folds` allows for specifying the number of training/test subsets to split the dataset in. In order to perform the classical leave-one-year-out cross-validation schema, `folds` should equal the total number of years encompassing the full training period (e.g.

`folds=list(1979:2008)`). The way the different subsamples are split is controlled by the argument `type`, providing fine control on how the random sampling is performed.

Here, in order to replicate the VALUE experimental Framework, a 5-fold cross-validation scheme is considered, each fold containing consecutive years for the total period 1979–2008 (Gutiérrez et al., 2019). The function `downscaleCV` thus performs the downscaling for each of the independent folds and reconstructs the entire time-series for the full period analyzed.

```
folds <- list(1979:1984, 1985:1990, 1991:1996,
                    1997:2002, 2003:2008)
```

The details for configuring the cross-validation of the methods in Table 2 are given throughout the following subsections:

### 3.3.2 Configuration of Method M1

Method M1 uses spatial predictors only. In particular, the (non rotated, combined) PCs explaining the 95% of total variance
are retained. As in the rest of methods, all the predictor variables are included to compute the PCs. The following argument list controls how the principal component analysis is carried-out, being internally passed to the function `prinComp` of package `transformeR`:

```
spatial.pars.M1 <- list(which.combine = vars,
                        v.exp = .95,
rot = FALSE)
```

As no other type of predictors (global and/or local) are used in the M1 configuration, the default values (`NULL`) assumed by `downscaleCV` are applied. However, for clarity, here we explicitly indicate these defaults in the command calls. As the internal object containing the PCA information bears all the data inside (inclusing PCs independently calculated for each variable), the argument `combined.only` serves to discard all the unnecessary information. Therefore, with this simple
specifications the cross-validation for method M1 is ready to be launched:

```
M1cv.bin <- downscaleCV(x = x, y = y_bin, method = "GLM",
         family = binomial(link = "logit"),
         folds = folds,
         prepareData.args = list(global.vars = NULL,
local.predictors = NULL,
                         spatial.predictors = spatial.pars.M1,
                         combined.only = TRUE))
```

In the logistic regression model, `downscaleCV` returns a multigrid with two output prediction grids, storing the variables *prob* and *bin*. The first contains the grid probability of rain for every day and the second is a binary prediction indicating
whether it rained or not. Thus, in this case the binary output is retained, using `subsetGrid` along the 'var' dimension:

```
M1cv.bin <- subsetGrid(M1cv.bin, var = "bin")
```

Next, the precipitation amount model is tested. Note that the *log* link function used in this case can't deal with zeroes in the data for fitting the model. Following the VALUE criterion, here a minimum threshold of 1 mm (`threshold = 1`, `condition = "GE"`, i.e., Greater or Equal) is considered:

```
M1cv.cont <- downscaleCV(x = x, y = y, method = "GLM",
                     family = Gamma(link = "log"),
                     condition = "GE", threshold = 1,
                     folds = folds,
                     prepareData.args = list(global.vars = NULL,
local.predictors = NULL,
                              spatial.predictors = spatial.pars.M1,
                              combined.only = TRUE))
```

The continuous and binary predictions are now multiplied using the `gridArithmetics` function from `transformeR`, so the precipitation frequency is adjusted and the final precipitation predictions are obtained:

```
M1cv <- gridArithmetics(M1cv.bin, M1cv.cont, operator = "*")
```

The final results stored in the `M1cv` grid can be easily handled for further analysis, as it will be later shown in Sec. 3.3.9 during method validation. As an example of a common check operation, here the (monthly accumulated and spatially averaged) predicted and observed time series are displayed using `temporalPlot` from package `visualizeR` (Fig. 3):

```
      aggr.pars <- list(FUN = "sum", na.rm = TRUE)
pred.M1 <- aggregateGrid(M1cv, aggr.m = aggr.pars)
      obs <- aggregateGrid(y, aggr.m = aggr.pars)
      temporalPlot(pred.M1, obs) ## Generates Fig. 3
```

### 3.3.3   Configuration of method M2

Unlike M1, in M2 the PCs are independently calculated for each variable, instead of considering one single matrix formed by
all joined (combined) variables. To specify this PCA configuration, the spatial predictor parameter list is modified accordingly, by setting `which.combine = NULL`.

```
      spatial.pars.M2 <- list(which.combine = NULL, v.exp = .95)
```

Note that the rotation argument is here omitted, as it is unused by default. This list of PCA arguments is passed to the
`spatial.predictor` argument. The rest of the code to launch the cross-validation for M2 is identical to M1.

### 3.3.4   Configuration of method M3

Method M3 uses local predictors only. In this case, the first closest neighbour to the predictand location (`n=1`) is used considering all the predictor variables (as returned by the helper `getVarNames(x)`). The local parameters is list is next defined:

```
      local.pars.M3 <- list(n = 1, vars = getVarNames(x))
```

In addition, the scaling parameters control the raw predictor standardization. Within the cross-validation setup, standardiza-
tion is undertaken after data splitting. In this particular case (5 folds), the 4 folds forming the training set are jointly standard-

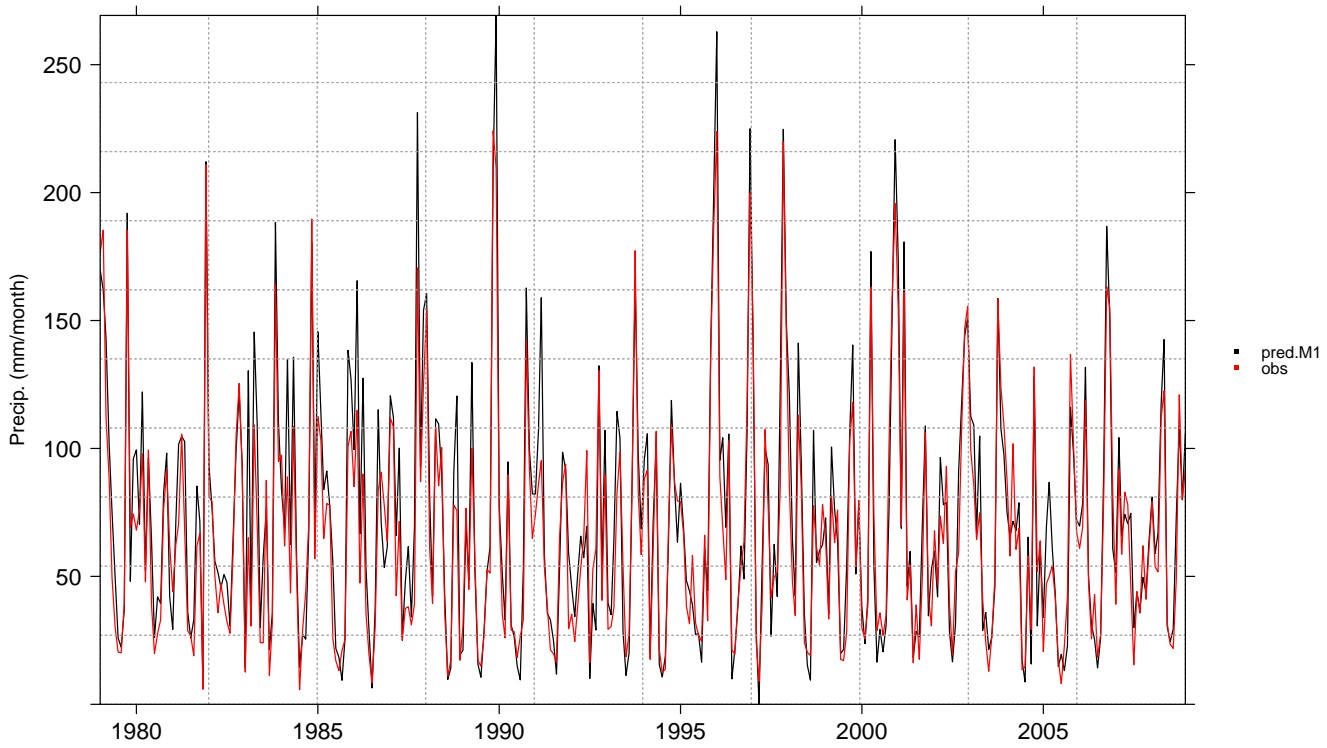

**Figure 3.** Cross-validated predictions of monthly accumulated precipitation by the method M1 (black), plotted against the corresponding observations (red). Both time series have been spatially aggregated considering the 11 stations within the Iberian subdomain.

ized. Then, its mean and variance is used for the standardization of the remaining fold (i.e., the test set). Therefore, the standardization parameters are passed to function `downscaleCV` as a list of arguments controlling the scaling (`scaling.pars` object; these parameters are passed internally to the function `scaleGrid`):

```
scaling.pars <- list(type = "standardize",
                     spatial.frame = "gridbox")
```

The next steps are similar to those already shown for M1. For clarity, the precipitation amount M3 model is next shown (the binary logistic model of occurrence would use a similar configuration, but changing the model family, as previously shown).

```
M3cv.cont <- downscaleCV(x = x, y = y, method = "GLM",
               family = Gamma(link = "log"),
               condition = "GE", threshold = 1,
               folds = folds,
               scaleGrid.args = scaling.pars,
```

```
        prepareData.args = list(global.vars = NULL,
                      local.predictors = local.pars.M3,
                      spatial.predictors = NULL))
```

### 3.3.5 Configuration of method M4

Method M4 is similar to M3, but using the 4 closest predictor gridboxes, instead of just one. Thus, the local predictor parameters are slightly modified, by setting `n = 4`:

```
local.pars.M4 <- list(n = 4, vars = vars)
```

### 3.3.6 Configuration of method M5

Method M5 uses raw (standardized) spatial predictor fields, instead of PCA-transformed ones. The standardization is performed by centering every gridbox with respect to the overall spatial mean, in order to preserve the spatial consistency of the standardized field. To account for this particularity, the scaling parameters are modified accordingly, via the argument `spatial.frame = "field"`, which is internally passed to `scaleGrid`.

```
scaling.pars.M5 <- list(type = "standardize",
                        spatial.frame = "field")
```

In this case, the method for model training is set to analogs. Other specific arguments for analog method tuning are used, for instance, the number of analogs considered (1 in this case):

```
M5cv <- downscaleCV(x = x, y = y,
          method = "analogs", n.analogs = 1,
          folds = folds,
          scaleGrid.args = scaling.pars.M5,
          prepareData.args = list(global.vars = vars,
                      local.predictors = NULL,
                      spatial.predictors = NULL))
```

### 3.3.7 Configuration of method M6

The parameters used for predictor configuration in method M6 (combined PCs explaining 95% of total variance) are similar to method M1. Thus, the previously defined parameter list `spatial.pars.M1` is reused here:

```
M6cv <- downscaleCV(x = x, y = y,
          method = "analogs", n.analogs = 1,
          folds = folds,
          prepareData.args = list(global.vars = NULL,
                  local.predictors = NULL,
                  spatial.predictors = spatial.pars.M1,
                  combined.only = TRUE))
```

### 3.3.8 Configuration of method M7

Similarly, method M7 uses identical spatial parameters as previously used for method M2 (parameter list `spatial.pars.M2`), being the rest of the code similar to M6, but setting `combined.only = FALSE`, as independent PCs are used instead of the combined one.

### 3.3.9 Validation

Once the cross-validated predictions for the methods M1 to M7 are generated, their evaluation is undertaken following the systematic approach of the VALUE framework. For brevity, in this example the code of only two example indices is shown: Relative wet-day frequency (R01) and Simple Day Intensity Index (SDII). The evaluation considering a more complete set of 9 validation indices is included in the supplementary notebook to this paper (see the Code and Data availability Section), following the subset of measures used in the VALUE synthesis paper by Gutiérrez et al. (2019). Alternatively, a complete list of indices and measures and their definitions is available in a dedicated section in the VALUE Validation Portal (http://www.value-cost.eu/validationportal/app/#!indices). It is also possible to have a quick overview of the available indices and measures within the R session by using the helper functions `VALUE::show.indices()` and `VALUE::show.measures()`.

To apply them, the package `climate4R.value`, already introduced in Sec. 2.3, is used. The function `valueMeasure` is the workhorse for computing all the measures defined by the VALUE Framework. For example, to compute the ratio of the frequency of wet days (VALUE code R01) for a given cross-validated method (M6 in this example), the parameters `measure.code="ratio"` and `index.code="R01"` are given:.

```
R01.ratio <- valueMeasure(y, x = M6cv,
                          measure.code = "ratio",
                          index.code = "R01")$Measure
```

A spatial plot helps to identify at a glance at which locations the frequency of wet days is under/over (red/blue) estimated by method M6 (Fig. 4):

```
## Generates Fig. 4:
spatialPlot(R01.ratio, backdrop.theme = "countries")
```

Following with this example and using the 9 indices used in the synthesis of the VALUE experiment results (Maraun et al., 2019b), and considering the battery of all methods, M1 to M7, a summary of the validation is presented in Fig. 5. The figure has been generated with the function `violinPlot` from package `visualizeR`, as illustrated step by step in the companion paper notebook (see the Code and Data availability Section). Violins are in essence a combination of a box plot and a kernel density plot. Boxplots are a standard tool for inspecting the distribution of data most users are familiar with, but that lack basic information when data are not normally distributed. Density plots are more useful when it comes to compare how different datasets are distributed. For this reason, violin plots incorporate the information of kernel density plots in a boxplot-like representation, and are particularly useful to detect bimodalities or departures from normal distribution of the

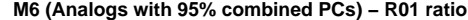

**M6 (Analogs with 95% combined PCs) – R01 ratio**

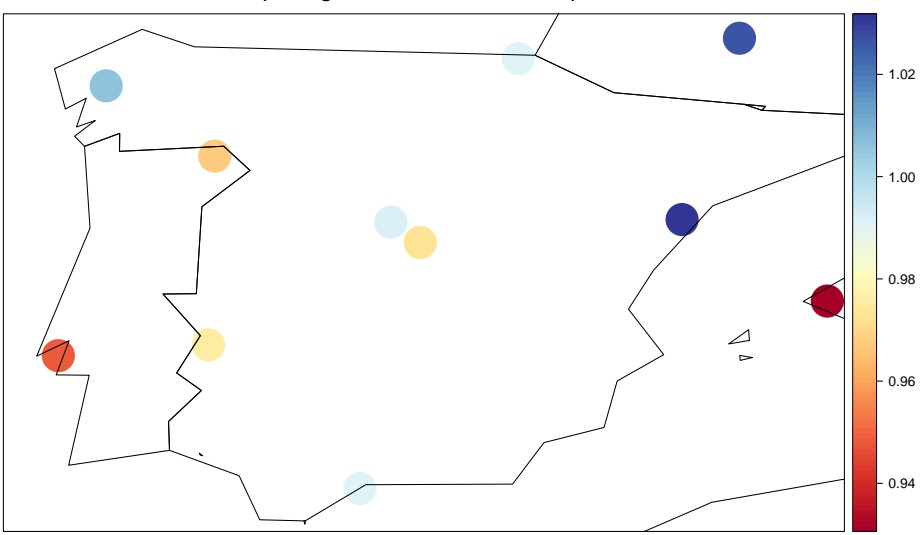

**Figure 4.** Cross-validation results obtained by method M6, considering the ratio (predicted/observed) of the frequency of wet days (VALUE index code R01, Table 1).

data, intuitively depicted by the shape of the violins. The violins are internally produced by the package `lattice` (Sarkar, 2008) via the panel function `panel.violin` to which the interested reader is referred for further details on violin plot design

570 and options. All the optional graphical parameters of the original `panel.violin` function can be conveniently passed to the wrapper `violinPlot` of package `visualizeR`. In the following, the violin plots shown display how the different validation measures are distributed across locations.

## 4 Contribution to VALUE: Further results

The methods M1* and M6* (see Table 2) contributed to the VALUE intercomparison experiment (see methods GLM-DET and

575 ANALOGS in Gutiérrez et al., 2019, Table 3) over the whole European domain, exhibiting a good overall performance. In this section we investigate the potential added value of including local information to these methods. To this aim, the VALUE M1* and M6* configurations are modified by including local information from neighbouring predictor gridboxes (these configurations are labelled as M1-L and M6-L respectively, Table 2). The M1-L and M6-L models are trained considering the whole pan-European domain, instead of each subregion independently, taking advantage of the incorporation of the local information

580 at each predictand location, thus disregarding the intermediate step of subsetting across subregions prior to model calibration. The experiment seeks to explore if the more straightforward local predictor approach (M1-L and M6-L) is competitive against the corresponding M1 and M6 VALUE methods when trained with one single, pan-European domain, instead of using the

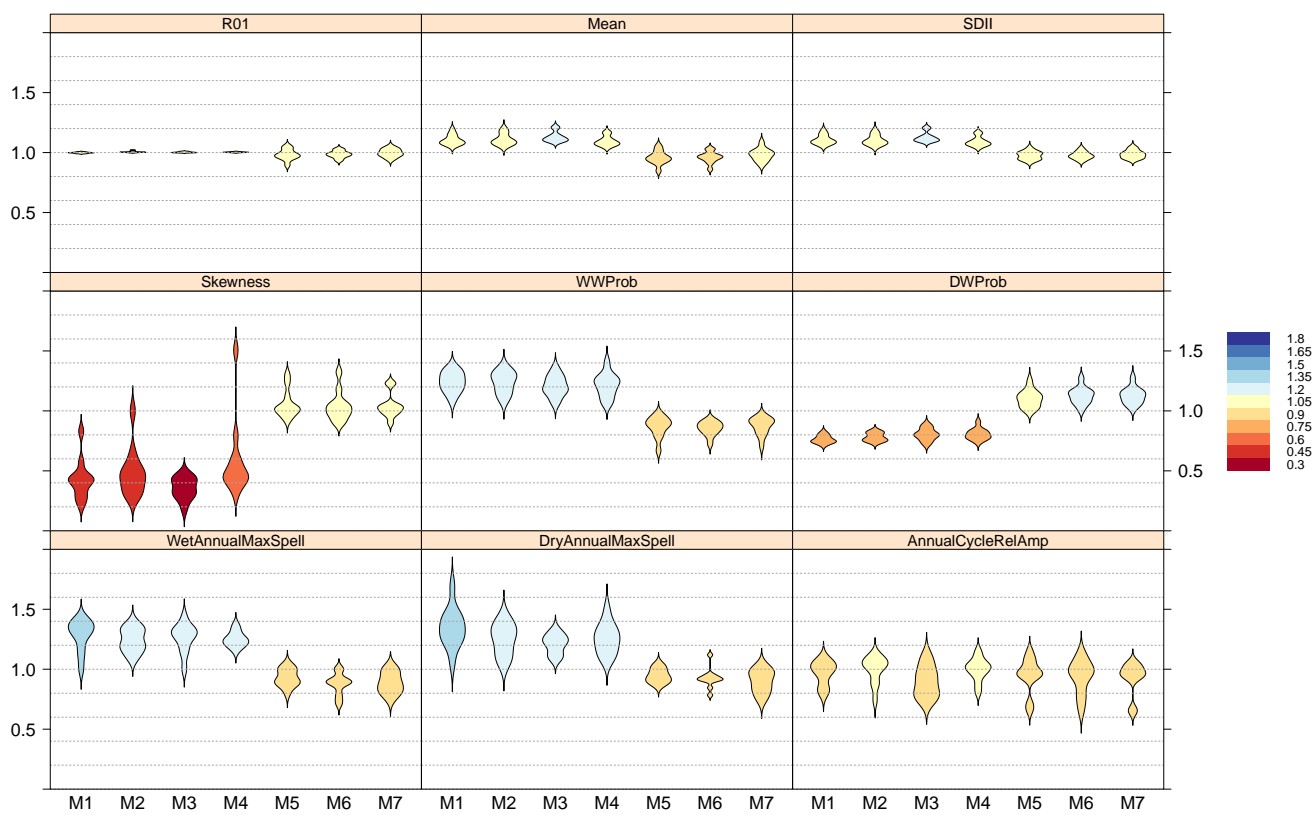

**Figure 5.** Cross-validation results obtained by the 7 methods tested (M1 to M7, Table 2) according to the core set of validation indices defined in the VALUE intercomparison experiment, considering the subset of the Iberian Peninsula stations ($n$=11). The colour bar indicates the mean ratio (predicted/observed) measure calculated for each validation index (Table 1).

VALUE subregional division, which poses a clear advantage from the user point of view as it does not require testing different spatial domains and the definition of subregions in large downscaling experiments.

585      Throughout this section, the pan-European experiment is launched and its results presented. Note that now the predictor *multigrid* corresponds to the whole European domain and the predictand contains the full set of VALUE stations (Fig. 2). The procedure for loading these data is identical to the one already presented in Sections 3.2.1 and 3.2.2, but considering the European domain. This is achieved by introducing the bounding box defined by the arguments `lonLim = c(-10,32)` and `latLim = c(36,72)` in the call to the `loadGridData` function. These arguments can be omitted in the case of the

590 station data load, since *all* the available stations are requested in this case. The full code used in this step is detailed in the companion paper notebook (see the Code and Data Availability section).

## 4.1 Method Intercomparison experiment

The configuration of predictors is indicated through the parameter lists, as shown throughout Sections 3.3.2 to 3.3.8. In the case of method M1-L, local predictors considering the first nearest gridbox are included in the M1 configuration (Table 2).:

```
M1.L <- list(local.predictors = list(n = 1, vars = vars),
                    spatial.predictors = list(v.exp = .95,
                                              which.combine = vars))
```

Unlike M6, the M6-L configuration considers local predictors only instead of PCs. In this case, the local domain window is wider than for M1-L, including the 25 closest gridboxes instead of just one:

```
M6.L <- list(local.predictors = list(n = 25, vars = vars))
```

Next, the cross validation is launched using `downscaleCV`. M1-L corresponds to the GLM method (thus requiring the two models for occurrence and amount), while M6-L is an analog method. After this, the validation is undertaken using `valueMeasure`. PP methods in general build on a synchronous daily link established between predictor(s) and predictand in the training phase (Sec. 2). The strength of this link indicates the local variability explained by the method as a function of the large-scale predictors. In order to provide a quick diagnostic of this strength for the different methods, and at the same time to illustrate a diversity of validation methods, in this case correlation, root mean square error and variance ratio are chosen as validation measures in the validation (Table 1). The validation results are displayed in Fig. 6. For brevity, the code performing the validation of the pan-European experiment is not repeated here (this is similar to what it has been already shown in Sections 3.3.2 to 3.3.8). The validation results indicate that the local predictor counterparts of the original VALUE methods M1 and M6 are competitive (the reach very similar or slightly better performance in all cases). Hence, the M1-L and M6-L method configurations will be used in Sec. 4.2 to produce the future precipitation projections for Europe, provided their more straightforward application as they do not need to be applied independently for each subregion. While the GLM method improves the correlation between predicted and observed series, the Analog approach does a better job in preserving the observed variability.

## 4.2 Future downscaled projections

In this section, the calibrated SD models are used to downscale GCM future climate projections from the CMIP5 EC-EARTH model (Sec. 3.1.3). Before generating the model predictions (Sec. 4.2.2), the perfect-prog assumption regarding the good representation by the GCM of the reanalysis predictors is assessed in Sec. 4.2.1,.

### 4.2.1 Assessing the GCM representation of the predictors

As indicated in Sec. 2.1, PP model predictions are built under the assumption that the GCM is able to adequately reproduce the predictors taken from the reanalysis. Here, this question is addressed through the evaluation of the distributional similarity between the predictor variables, as represented by the EC-EARTH model in the historical simulation, and the ERA-Interim

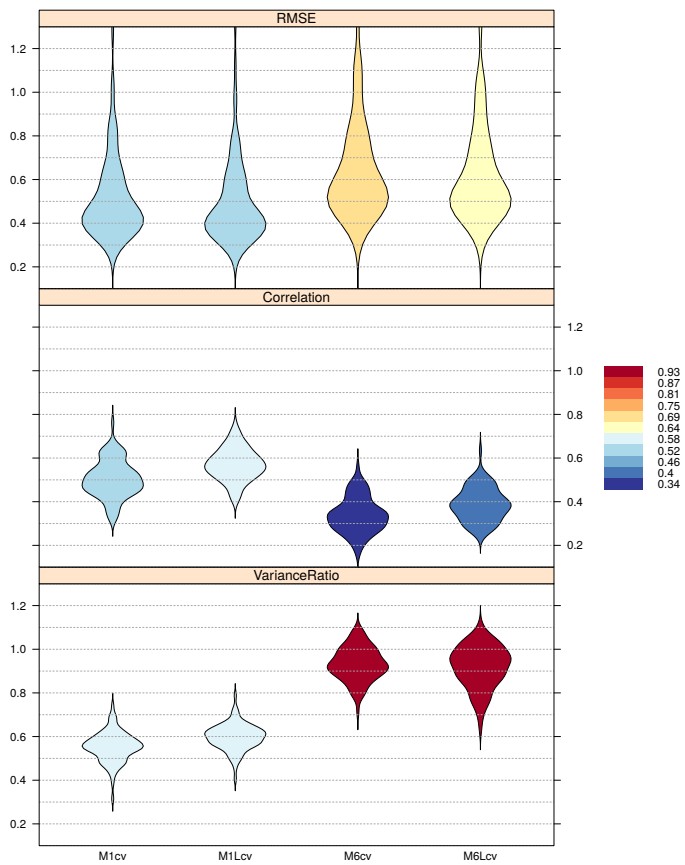

**Figure 6.** Cross-validation results obtained by the 4 methods tested (M1, M1-L, M6, and M6-L, Table 2) in the pan-European experiment ($n$=86 stations), according to three selected validation measures (Spearman correlation, RMSE and Variance ratio, see Table 1). The colour bar indicates the mean value of each measure. A factor of 0.1 has been applied to RMSE in order to attain the same order of magnitude in the Y-axis for all the validation measures.

reanalysis. To this aim, the two-sample Kolmogorov-Smirnov test is used, included in the set of validation measures of the VALUE framework and thus implemented in the VALUE package. The KS test is a non-parametric statistical hypothesis test

for checking the null hypothesis ($H_0$) that two candidate datasets come from the same underlying theoretical distribution. The statistic is bounded between 0 and 1, indicating the lower values a greater distributional similarity. The KS test is first applied to the EC-EARTH and ERA-Interim reanalysis time series at a grid box basis, considering the original continuous daily time series for their common period 1979-2005. In order to isolate distributional dissimilarities due to errors in the first- and second-order moments, we also consider anomalies and standardized anomalies (the latter being used as actual predictors in the SD

models). The anomalies are calculated by removing the overall grid-box mean to each daily value, and in the case of the standardized anomalies, we additionally divide by the seasonal standard deviation. Due to the strong serial correlation present in the daily time series, the test is prone to inflation of type-1 error, that is, rejecting the null hypothesis of equal distributions

|        | Min. | 1st Qu. | Median | Mean | 3rd Qu. | Max. | sd |
|--------|------|---------|--------|------|---------|------|-----|
| RMSE(×0.1) | | | | | | | |
| M1cv  | 0.27 | 0.39 | 0.45 | 0.52 | 0.60 | 1.41 | 0.20 |
| M1Lcv | 0.25 | 0.37 | 0.43 | 0.49 | 0.58 | 1.33 | 0.19 |
| M6cv  | 0.33 | 0.49 | 0.57 | 0.67 | 0.78 | 1.96 | 0.28 |
| M6Lcv | 0.32 | 0.47 | 0.55 | 0.64 | 0.74 | 1.74 | 0.26 |
| Correlation | | | | | | | |
| M1cv  | 0.32 | 0.45 | 0.50 | 0.50 | 0.55 | 0.76 | 0.09 |
| M1Lcv | 0.40 | 0.52 | 0.56 | 0.57 | 0.62 | 0.76 | 0.07 |
| M6cv  | 0.16 | 0.28 | 0.34 | 0.34 | 0.39 | 0.56 | 0.08 |
| M6Lcv | 0.25 | 0.33 | 0.39 | 0.39 | 0.44 | 0.63 | 0.08 |
| Variance Ratio | | | | | | | |
| M1cv  | 0.32 | 0.52 | 0.55 | 0.55 | 0.59 | 0.74 | 0.07 |
| M1Lcv | 0.41 | 0.57 | 0.60 | 0.60 | 0.63 | 0.79 | 0.06 |
| M6cv  | 0.72 | 0.88 | 0.93 | 0.93 | 0.99 | 1.08 | 0.08 |
| M6Lcv | 0.64 | 0.86 | 0.94 | 0.92 | 0.98 | 1.10 | 0.10 |

**Table 3.** Validation results of the 4 methods tested in the pan-European experiment. The values presented (from left to right: minimum, first quartile, median, third quartile, maximum and standard deviation) correspond to the violin plots displayed in Fig. 6 ($n = 86$ stations). Note that, for consistency with Fig. 6, the RMSE results are multiplied by a factor of $0.1$ in order to attain a similar order of magnitude for the three validation measures considered. This is also indicated in the caption of Fig. 6.

when it is actually true. To this aim, an effective sample size correction has been applied to the data series to calculate the p-values (Wilks, 2006). The methodology follows the procedure described in Brands et al. (2012, 2013), implemented by the VALUE measure 'ts.ks.pval' (Table 1).

The distributions of GCM and reanalysis (Fig. 7) differ significantly when considering the raw time series, thus violating the assumptions of the PP hypothesis. Centering the data (i.e, zero mean time series) greatly alleviates this problem for most variables, excepting specific humidity at 500 mb ('hus@500'), and near-surface temperature ('tas', not shown here, but displayed in the paper notebook). Finally, data standardization improves the distributional similarity, attaining an optimal representation of all the GCM predictors but 'hus@500' over a few grid points in the Mediterranean.

The distributional similarity analysis is straightforward using the functions available in `climate4R`, already shown in the previous examples. For brevity, the code generating Fig. 7 is omitted here, and included with extended details and for all the predictor variables in the companion paper notebook (see the Code and Data Availability Section).

  – Data centering/standardization is performed directly using the function `scaleGrid`, using the appropriate argument
     values `type="center"`/`"standardize"` respectively.

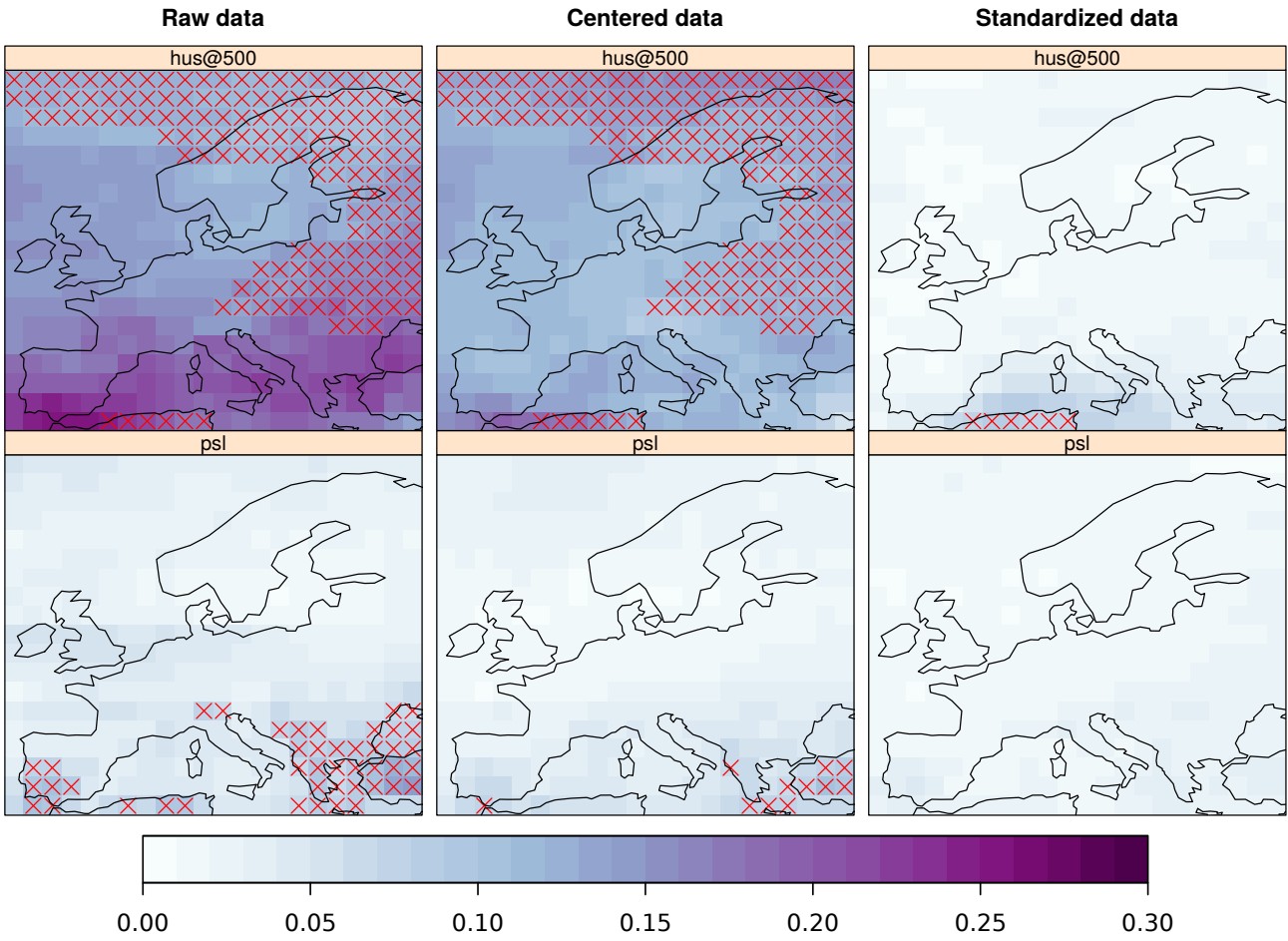

**Figure 7.** KS score maps, depicting the results of the two-sample KS test applied to the time series from the EC-EARTH GCM and ERA-Interim, considering the complete time series for the period 1979-2005. The results are displayed for two of the predictor variables (by rows), namely Specific humidity at 500 mb surface pressure height ("hus@500", badly represented by the GCM) and mean sea-level pressure ("psl", well represented by the GCM). The KS test results are displayed by columns, using, from left to right: the raw, the zero-mean (centered) and the zero-mean and unit variance (standardized) time series from both the reanalysis and the GCM. The grid boxes showing low p-values ($p < 0.05$) have been marked with a red cross, indicating significant differences in the distribution of both GCM and reanalysis time series.

– The KS-test is directly launched using the function `valueMeasure` from package `climate4R.VALUE`, and including the argument value `measure.code="ts.ks"` or `"ts.ks.pval"` for KS-score and its (corrected) p-value respectively.

– The KS score maps and the stippling based on their p-values are produced with the function `spatialPlot` from package `visualizeR`.

In conclusion, although not all predictors are equally well represented by the GCM, data standardization is able to cure the problem of distributional dissimilarities, even in the case of the worst represented variable, that is, specific humidity at 500 mb level.

### 4.2.2 Future SD model predictions

The final configuration of predictors for M1-L (stored in the `M1.L` list) and M6-L methods (`M6.L`) is directly passed to the function `prepareData`, whose output contains all the information required to undertake model training via the `downscaleTrain` function. In the following, the code for the analog method is presented. Note that for GLMs the code is similar, but taking into account occurrence and amount in separated models, as previously shown.

Unlike `downscaleCV`, than handles predictor standardization on a fold-by-fold basis (see Sec. 3.3.1 in the configuration
of method M3), predictor standardization need to be undertaken prior to passing the predictors to the function `prepareData`

```
# Standardization
x_scale <- scaleGrid(x, type = "standardize")
# Predictor config (M6-L method)
M6L <- prepareData(x_scale, y, local.predictors = M6.L)
# SD model training
model.M6L <- downscaleTrain(M6L, method = "analogs",
                                  n.analogs = 1)
```

After SD model calibration `downscalePredict` is the workhorse for downscaling. First of all, the GCM datasets required are obtained. As previously done with ERA-Interim, the EC-EARTH simulations are obtained from the `climate4R` UDG,
considering the same set of variables already used for training the models (Sec. 3.1.2). Again, the individual predictor fields are recursively loaded and stored in a `climate4R` *multigrid*.

```
historical.dataset <- "CMIP5_EC-EARTH_r12i1p1_historical"
grid.list <- lapply(variables, function(x) {
                loadGridData(dataset = historical.dataset,
var = x,
                                 lonLim = c(-10,32),
                                 latLim = c(36,72),
                                 years = 1979:2005)
    }
)
```

As done with the predictor set, the prediction dataset is also stored in as a *multigrid* object:

```
xh <- makeMultiGrid(grid.list)
```

An additional step entails regridding the GCM data onto the predictor grid prior to downscaling, in order to attain spatial consistency between the predictors and the new prediction data. This is done using the `interpGrid` function from
685 `transformeR`:

```
xh <- interpGrid(xh, new.coordinates = getGrid(x))
```

Identical steps are followed in order to load the future data from RCP8.5. Note that in this case, it suffices with replacing the URL pointing to the historical simulation dataset by the one of the future scenario chosen, in this case `dataset = "CMIP5_EC-EARTH_r12i1p1_rcp85"`. The *multigrid* object storing the future GCM data for prediction will be named xf.

Prior to model prediction, data harmonization is required. This step consists of rescaling the GCM data to conform to the mean and variance of the predictor set that was used to calibrate the model. Note that this step is achieved through two consecutive calls to `scaleGrid`:

```
xh <- scaleGrid(xh, base = xh, ref = x,
        type = "center",
        spatial.frame = "gridbox",
        time.frame = "monthly")
xh <- scaleGrid(xh, base = x, type = "standardize")
```

Again, an identical operation is undertaken with the future dataset, by just replacing xh by xf in the previous code chunk. Then, the function `prepareNewData` will undertake all the necessary data collocation operations, including spatial and temporal checks for consistency, leaving the data structure ready for prediction via `downscalePredict`. This step is performed equally for the historical and the future scenarios:

```
h_analog <- prepareNewData(newdata = xh, data.struc = M6L)
f_analog <- prepareNewData(newdata = xf, data.struc = M6L)
```

Finally, the predictions for both the historical and the future scenarios are done with `downscalePredict`. The function receives two arguments: i) `newdata`, where the pre-processed GCM predictors after `prepareNewData` are stored, and ii) `model`, which contains the model previously calibrated with `downscaleTrain`:

```
hist_ocu_glm <- downscalePredict(newdata = h_analog,
                                 model = model.M6L)
```

Once the downscaled future projections for historical and RCP 8.5 scenarios are produced using the methods M1-L (GLMs) and M6-L (Analogs), their respective predicted climate change signals (or "deltas") are displayed in Fig. 8 (the code to generate the figure is illustrated in the companion paper notebook, see the Code and Data availability Section). We also depict the downscaled climate change signals for the M1 and M6 configurations in order to evaluate whether the local-window approach alters the climate change signals. As illustrated in Fig.8, the projected relative changes in the climate signal of the R01 (first row) and SDII (second row) indices show minor differences among the configurations presented herein (i.e., M1-L and M6-L) and the VALUE methods (i.e., M1 and M6), showing that the uncertainty due to the SD method in the climate change signal (M1 –GLMs– *vs*. M6 –analogs–) is larger than that between global predictors/local window (M1/M6 *vs*. M1-L/M6-L respectively), in agreement with San-Martín et al. (2016). This result further supports the idea of replacing the VALUE subdomain approach by the adaptive window centered on each predictand location, allowing for a much more straightforward performance of large PP experiments encompassing large areas without the need of testing different subdomain configurations.

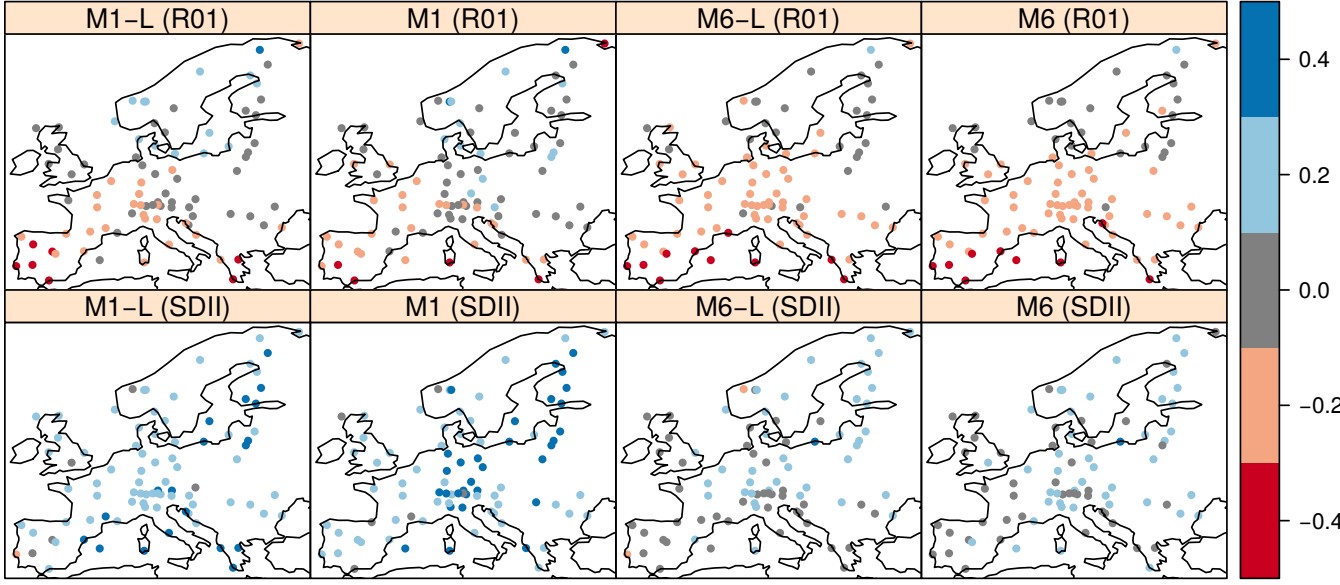

**Figure 8.** Relative delta change signals of the R01 and SDII precipitation indices (see Table 1) for the future period 2071–2100 (w.r.t. the baseline 1979–2005), obtained by the downscaled projections of the CMIP5 GCM EC-EARTH-r12i1p1, considering the RCP8.5 experiment. The SD methods used are M1-L, M1, M6 and M6-L (see Table 2).

## 5   Conclusions

The results obtained in the pan-European method intercomparison experiment (Sec. 4.1), indicate that the example SD methods contributing to the VALUE Experiment (GLMs and Analogs, first reproduced in Sec. 3.3.1), can be improved through the incorporation of local predictors, a novel feature brought by `downscaleR` that can help to avoid the burden of spatial domain screening. It has been shown that this method does not significantly alter the SD model results, neither in current climate validation, nor in regard with the projected anomalies. These results are of relevance for the development of the forthcoming EURO-CORDEX SD statistical downscaling scenarios, in which the VALUE activities have merged and will follow on, greatly facilitating the development of downscaling experiments over large areas, like the continental scale considered in this study. As in any other experiment, caution must be taken in order to ensure that the assumptions for perfect-prog applications are fulfilled, as shown here.

The experiments carried out throughout Sections 3.3 and 4 have served to the purpose of showcasing the most prominent features of the R package `downscaleR` and its integration in the `climate4R` framework, demonstrating its use in end-to-end applications. With this regard, `downscaleR` is a new tool implementing state-of-the-art SD techniques providing an extremely flexible interface to accomplish complex downscaling experiments. Critical aspects to be considered in any downscaling exercise, including domain definition, predictor configuration, perfect-prog hypothesis testing, model validation and intercomparison, can be achieved through the use of a few intuitive commands. Users of `downscaleR` can also benefit

from its direct integration within the comprehensive, well-consolidated VALUE framework for model evaluation. Furthermore, its full integration with `climate4R` brings to climate scientists and practitioners a unique comprehensive R-based framework for SD model development, including a cloud-computing facility, user-friendly data access to a large climate database and efficient solutions for data manipulation, visualization and analysis within one single computing environment.

*Code and data availability.* In order to promote transparency and research reproducibility, all the steps followed to generate the analyses shown in this paper (with extended details and additional information), are available in the companion Paper Notebook (repo version 0.1.4, https://doi.org/10.5281/zenodo.3567736):

– source file (R markdown): https://github.com/SantanderMetGroup/notebooks/blob/v0.1.4/2019_downscaleR_GMD.Rmd

– html file: https://github.com/SantanderMetGroup/notebooks/blob/v0.1.4/2019_downscaleR_GMD.html

– pdf file: https://github.com/SantanderMetGroup/notebooks/blob/v0.1.4/2019_downscaleR_GMD.pdf

The R software and all the packages required to reproduce the results are freely available as indicated in the paper notebook, where more specific details for installation and required versions are given.

– Name of the software: `downscaleR` (paper version: 3.1.0, https://www.doi.org/10.5281/zenodo.3277316)

– Developers: Authors of this paper

– Website: https://github.com/SantanderMetGroup/downscaleR

– Hardware Requirements: General-purpose computer

– Programming Languages: R

– Software Requirements: R version 3.1.0 or later.

## Appendix A: Computing times

### Method

The computing performance of the different downscaling experiments is analysed in this Appendix through the use of one indicator, the computing time, which measures the (user) time required to accomplish a certain task. Therefore, all timings presented in the following plots correspond to user (wall) times. The values shown are mean values after considering $n = 10$ experiment replicates in all cases. However, spread measures are not displayed given that their values are negligible, attaining all realizations very similar timings.

All timings presented have been measured using the R package `microbenchmark` (Mersmann, 2019), on a dedicated Ubuntu 16.04 LTS (64 bits), with 15.6 GiB memory and a multi-core CPU composed on 8 processing units Intel® Core™ i7-6700 of 3.40GHz. Further details on the R configuration are provided in the Session Information section of the companion paper notebook.

### Results

The computing times for the Iberia and Pan-European downscaling experiments are depicted in Fig.A1 and Fig.A2, respectively. A more detailed description of the process naming is indicated in Table A1. The different downscaling configurations are named according to Table 2, and match the nomenclature used in the companion paper notebook. As it can be seen, all the method families perform similarly, being the analogs approach in general significantly slower that GLMs, highlighting the computationally demanding task of analog search (methods M5-M7), that is significantly reduced when the dimensionality of the predictor set is reduced using PCs (M6 and M7). On the other hand, the use of local neighbors instead of PCs does not make a significant difference in computing times, as it can be seen from the intercomparison of GLM methods (M1 to M4, Fig. A1). As expected, downscaling the Pan-European domain (i.e., configurations M1L and M6L) leads to higher computational times in comparison with the Iberian downscaling experiment (see Fig. A2), especially in the analogs case, in which the analog search is computationally demanding due to the larger size of the Europe-wide predictor set. The comparison between training and testing times show that the most time-consuming sub-task is the preparation of the predictor and SD model training, in this order (Fig. A3), while prediction is much faster in general for all the methods.

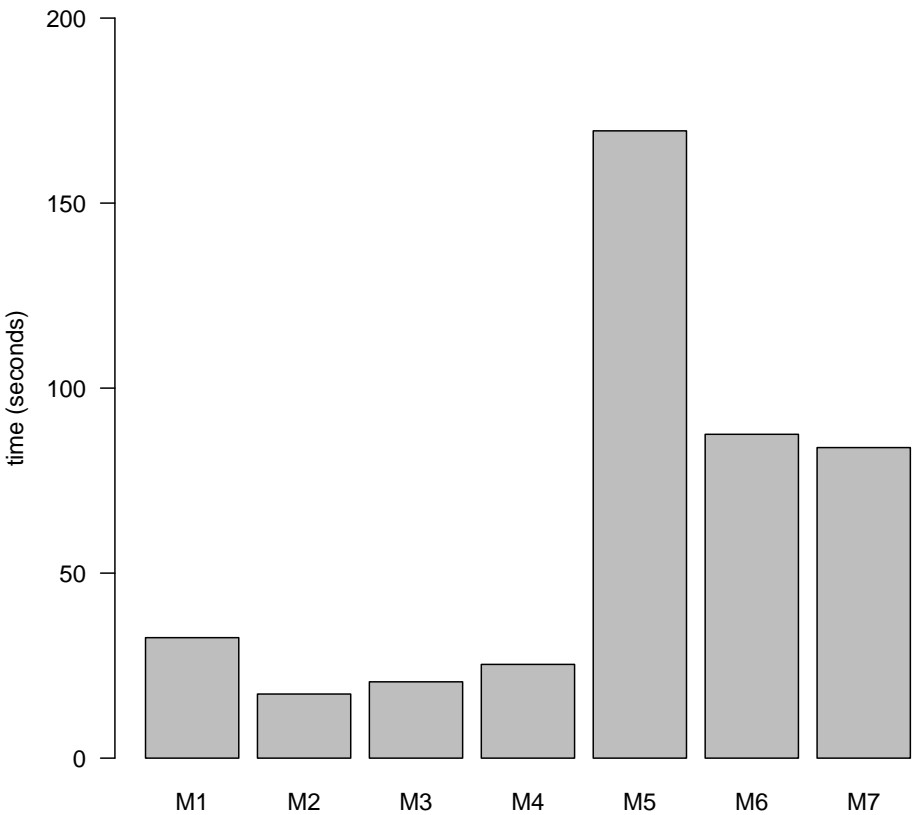

**Figure A1.** Cross-validation times required for the downscaling models developed in the Iberian experiment. The computational times of the generalized linear models configurations (see Table A1) includes both the downscaling of the occurrence and amount of precipitation, whereas for the analogs both aspects are downscaled simultaneously. More information about the configurations can be found in Tables A1 and 2, or in the companion paper notebook.

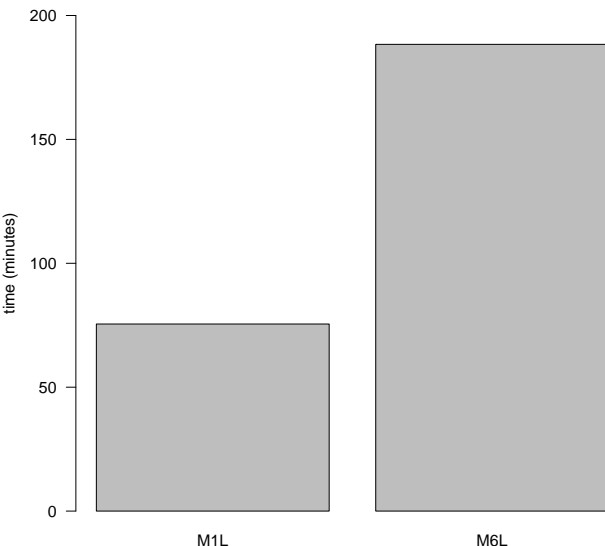

**Figure A2.** Cross-validation times required for the downscaling models developed in the Pan-European experiment. The computational times of the generalized linear models configurations (see Table A1) includes both the downscaling of the occurrence and amount of precipitation, whereas for the analogs both aspects are downscaled simultaneously. More information about the configurations can be found in Tables A1 and 2, or in the companion paper notebook.

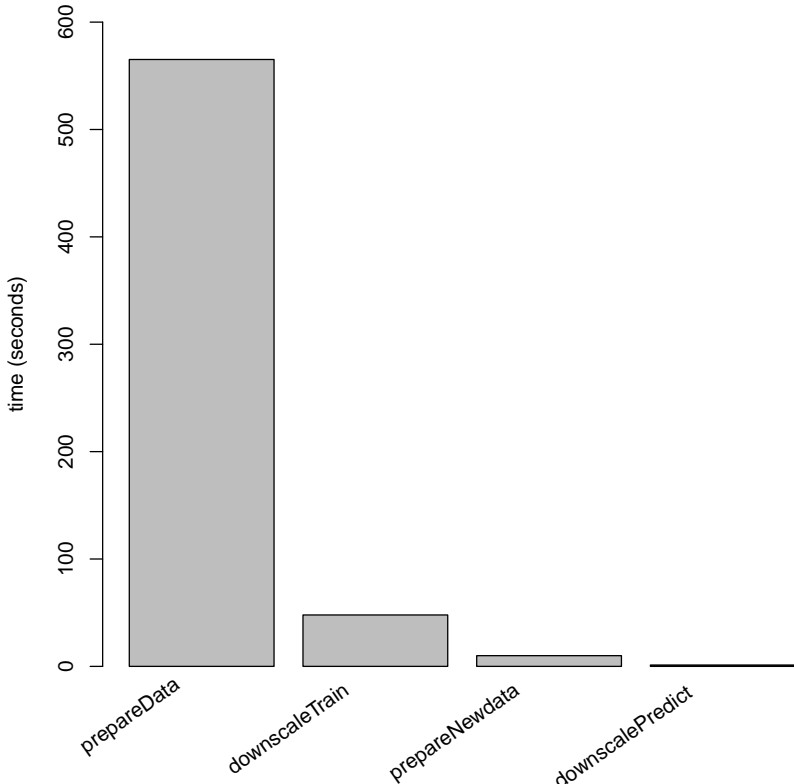

**Figure A3.** Computing times of a particular method (M1-L for precip. occurrence, Table A1) considering the European domain experiment ($n = 86$ stations, 1979-2008). The bulk of computing time is for predictor preparation, and model fitting, while the preparation of the new data and the predictions are relatively much faster.

| Configuration | Region | Operation | Method | Spatial features (PCs) | Local features |
|---|---|---|---|---|---|
| M1 | Iberia | Cross-validation | GLM | yes | no |
| M2 | Iberia | Cross-validation | GLM | yes | no |
| M3 | Iberia | Cross-validation | GLM | no | yes |
| M4 | Iberia | Cross-validation | GLM | no | yes |
| M5 | Iberia | Cross-validation | Analogs | no | no |
| M6 | Iberia | Cross-validation | Analogs | yes | no |
| M7 | Iberia | Cross-validation | Analogs | yes | no |
| M1L | Europe | Cross-validation | GLM | yes | yes |
| M6L | Europe | Cross-validation | Analogs | yes | yes |
| M1L (downscaleTrain) | Europe | Training | GLM | yes | yes |
| M1L (downscalePredict) | Europe | Testing | Analogs | yes | yes |

**Table A1.** A brief description of the nomenclature used in Figs. A1,A2 and A3, involving the predictor configuration (i.e., spatial and/or local features), the region, and the method (i.e., GLM or analogs). Also detailed description of these configurations can be found in Table 2.

*Author contributions.* All the authors have contributed to the downscaleR package development. J.B., J.B.M and J.M.G. wrote an initial version of the manuscript and designed the illustrative experiments. J.B and J.B.M prepared the companion Paper Notebook and the analyses shown in the Appendix.

*Competing interests.* The authors declare no competing interests.

*Acknowledgements.* We thank the European Union Cooperation in Science and Technology (EU COST) Action ES1102 VALUE (www. value-cost.eu) for making publicly available the data used in this article and the tools implementing the comprehensive set of measures and indices for the validation of downscaling results. We also thank the THREDDS Data Server (TDS) software developed by UCAR/Unidata (http://doi.org/10.5065/D6N014KG) and to all R developers and its supporting community for providing free software facilitating open science. We acknowledge the World Climate Research Program's Working Group on Coupled Modelling, which is responsible for CMIP, and 790 we thank the EC-EARTH Consortium for producing and making available their model output used in this paper. For CMIP the U.S. Department of Energy's Program for Climate Model Diagnosis and Intercomparison provides coordinating support and led development of software infrastructure in partnership with the Global Organization for Earth System Science Portals. The authors acknowledge partial funding from MULTI-SDM project (MINECO/FEDER, CGL2015- 66583-R). The first author acknowledges funding from the Project INDECIS, part of European Research Area for Climate Services Consortium (ERA4CS) with co-funding by the European Union Grant 690462. We are very 795 grateful to the two anonymous referees participating in the interactive discussion for their insightful comments, helping us to considerably improve the original manuscript.

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
