# Peer review of "Statistical downscaling with the downscaleR package (v3.1.0): Contribution to the VALUE intercomparison experiment"

_Geoscientific Model Development, 2019_

## Short Comment (SC1) · 5 Nov 2019

Dear authors,

in my role as Executive editor of GMD, I would like to bring to your attention our Editorial version 1.2:

https://www.geosci-model-dev.net/12/2215/2019/

This highlights some requirements of papers published in GMD, which is also available on the GMD website in the 'Manuscript Types' section: http://www.geoscientific-model-development.net/submission/manuscript_types.html

[Figure]

In particular, please note that for your paper, the following requirement has not been met in the Discussions paper:

- "The main paper must give the model name and version number (or other unique identifier) in the title."

Please provide the version number of the downscaleR package in the title of your revised manuscript.

Additionally, please note, that GMD is encouraging authors to provide a persistent access to the exact version of the source code used for the model version presented in the paper. As explained in https://www.geoscientific-model-development.net/about/manuscript_types.html the preferred reference to this release is through the use of a DOI which then can be cited in the paper. For projects in GitHub a DOI for a released code version can easily be created using Zenodo, see https://guides.github.com/activities/citable-code/ for details.
Please note, that a GitHub is not necessarily a permanent archive !

Yours, Astrid Kerkweg

---

## Author Comment (AC1) · 5 Nov 2019

Dear Editor,

Thanks for your feedback on the article. We are aware of the requirement for a unique identifier of the software described in the paper. As a result, we have deposited our software in the Zenodo repository, obtaining a persistent DOI. This is the DOI of the software version (downscaleR v1.3.0) used in this article:

https://doi.org/10.5281/zenodo.3277316

We will include this information in the new revised version of the manuscript. Also, the
version will be clearly indicated in the title.

Your sincerely,

J. Bedia, on behalf of the authors

---

## Referee Comment (RC1) · Anonymous Referee #1 · 8 Nov 2019

A step commonly carried out when assessing the 'quality' or 'value' of climate data is the comparison with observed data, normally applying a downscaling step. This paper presents a reproducible R-based workflow in the context of the COST action VALUE. The paper presents a workflow (also shared as R Markdown notebook) which start with data loading to the visualisation of the results. In this workflow the authors compare different downscaling techniques.

I have a few comments here that I think would improve the submitted paper:

1. In the Section 4.1 the authors might add some numbers to Figure 6 (even a separate table) showing average (possibly also std or quantiles) values of RMSE, Correlation

and variance ratio. Comparing M1, M6 and their -L version graphically is not easy. 2. Again in Figure 6 I don't understand the meaning of 'A factor of 0.1 has been applied to RMSE for better comparability of results.', why not leaving the original values? 3. The authors should say something on the computation time needed for the experiments described in the Figure. 4. How the developed package is able to deal with large datasets (10-100-500GB)? Is there any support to larger-than-memory computing (e.g. Python Dask)? 5. Can the authors say something about the importance of choosing the right domain to compute the EOF? Sometimes the results can be very sensitive to the choice of the domain.

---

## Referee Comment (RC2) · Anonymous Referee #2 · 24 Nov 2019

The authors present the R package downscaleR. In principle this is a very useful contribution and worth publishing in GMD. But before publication I ask the authors to address the following major issues, plus a series of minor but still important ones.

Major issues

1. In section 4 the authors consider a pan-European setting, and explore whether models using a European predictor domain with additional local predictors perform equally well as the corresponding models with predictors defined on regional domains as used in VALUE. If these models would indeed perform well, this would mean a substantial simplification, e.g., for large-scale ESD applications such as in EURO-CORDEX. I am

afraid, however, that the reasoning is not quite stringent. The validation is based on ERA-Interim predictors, which should well represent local predictors given that local observations have been assimilated. In a GCM context, these local predictors may not fulfill the perfect prog condition, i.e., they may not be bias free. If this were the case, the GCM-based projections could be substantially biased, and the use of local predictors were not permitted. In fact, biases may also affect the climate change signal. I therefore ask the authors to test the PP assumption: first, they should use the historical simulations of their GCM-predictor experiment and check the perfect prog assumption. And second, they should investigate whether the climate change signals simulated by the local implementations differs from those of the VALUE implementations. If the PP assumption was not fulfilled, and/or if the climate change signal was modified, the authors should change their conclusions correspondingly. Even in a positive result, the authors should mention that care is required for the reasons given above.

2. I am wondering how downscaleR is placed relative to ESMValTool. This is a widely used tool mainly (but not exclusivel) in the GCM community, and it should be possible to combine analyses and results from the different tools. It would be disappointing if the two packages would not be compatible (beyond the exchange of NetCDF files), so a discussion is absolutely necessary, and compatibility very much desired.

3. The conclusions are quite weak. I would really appreciate if the authors could discuss what the purpose of the package is, and where it sits in the wide landscape of downscaling and evaluation tools in climate sciences, and what the specific advantages are. This has been touched in the introduction, but here it should be referred back, and some substantial statements should be made.

Minor issues:

In general, some minor grammatical errors (e.g. l 192 "analogs performance") need to be corrected.

l 5: VALUE is a network, not a project. You might also call it an initiative.

l 25: "are not suitable" This is not always true. Please replace by "are often not suitable"

l 32: "SD" here you could refer to a recent review or introductory text, e.g., Maraun & Widmann, CUP, 2018.

l 40: "It must be noted" is a zero phrase. Start with "SD techniques are..."

l 45: Here it would be fair to cite Barsugli et al., EOS, 2013.

l 55: Here it would be useful to cite the synthesis article, Maraun et al., IJC, 2019, highlighting that this article aims at giving an overall assessment of relative merits and limitations.

l 66: "It is worth mentioning here": Again, a zero phrase. You could rather state "This toolbox complements/adds to other existing tools..."

l 106: somewhere in the introduction you should mention ECMValTool

l 113: here you should really also refer to Maraun & Widmann, CUP, 2018. It is the most comprehensive discussion of the two approaches in a climate change context.

l 119: no - the term "perfect" refers to the assumption that the predictors are bias free. In particular in weather forecasting, also for MOS the day-to-day correspondence is given. For the MOS discussion you should make clear that the limitation of having homogeneous predictor-predictand relationships applies only in a climate context. This is also the reason why MOS in climate research is typically just bias correction. In weather forecasting, you are as free as in PP.

l 130: you may consider presenting the updated assumptions formulated by Maraun & Widmann, CUP, 2019. They are more precise and include the often neglected requirement that the model structure should be applicable.

l 169: you may consider to add a comment that often predictors are proxies for physical processes, which is a main reason for nonstationarities in the predictor/predictand relationship, as amply discussed in Maraun & Widmann, CUP, 2019. In this context, you

should mention that predictor selection and the training of transfer functions are carried out on short term variability in present climate, whereas the aim is typically to simulate long term changes of short term variability (same reference, and Huth, J. Clim., 2004)

l 194: it should be pointed out that this is true only for analog methods, which use the same sequence of analogs for different locations. Otherwise spatial coherence is underestimated. This has been demonstrated by the cited Widmann et al., IJC, 2019.

l 196: this statement could be formulated much stronger. I am not aware of any region in the world, where climate change will be so moderate, that the analog method still applies in the far future, when temperature and directly related variables are considered.

l 205: somewhere you should mention that the main advantage of GLMs is to simulate (non-normal) variance not explained by the predictors (e.g., von Storch, J. Climate, 2000, although, strangely, not all models make use of that).

Fig 5: the violin plot needs some explanation. It is not quite clear what the distribution shows. Densities across stations? Is theres some kernel smoothing applied? Also: is this an annual analysis? The same holds for the following figures as well.

---

## Author Comment (AC2) · 9 Dec 2019

A step commonly carried out when assessing the 'quality' or 'value' of climate data is the comparison with observed data, normally applying a downscaling step. This paper presents a reproducible R-based workflow in the context of the COST action VALUE. The paper presents a workflow (also shared as R Markdown notebook) which start with data loading to the visualisation of the results. In this workflow the authors compare different downscaling techniques.

I have a few comments here that I think would improve the submitted paper:

1. In the Section 4.1 the authors might add some numbers to Figure 6 (even a separate table) showing average (possibly also std or quantiles) values of RMSE, Correlation and variance ratio. Comparing M1, M6 and their -L version graphically is not easy.

We have included the numbers corresponding to the validation of the pan-european experiment in the new Table 3 of the revised manuscript version.

2. Again in Figure 6 I don't understand the meaning of 'A factor of 0.1 has been applied to RMSE for better comparability of results.', why not leaving the original values?

We decided to apply a scaling factor of 0.1 to the RMSE values in order to make their magnitude comparable to that of the other validation measures, so they can be visually compared in the same plot. We have replace the caption indicating that "[...] The colour bar indicates the mean value of each measure. A factor of 0.1 has been applied to RMSE in order to attain the same order of magnitude in the Y-axis for all the validation measures", hoping that it is now more clear.

3. The authors should say something on the computation time needed for the experiments described in the Figure.

We have included a new section in Appendix 1 devoted to a more detailed analysis of computing times. Please find attached some figures to be included in the new revised version of the manuscript addressing the efficiency, in terms of computing (user) times, of the different downscaling methods. As an example, Fig. 1 shows the computing times required to accomplish each of the methods used in the Iberian Peninsula experiment. A more detailed discussion of these results and additional figures/tables are included in the revised version of the manuscript.

[Figure]

*Figure 1. Cross-validation times required for the downscaling models developed in the Iberian experiment. The computational times of the generalized linear models configurations (see Table A1) includes both the downscaling of the occurrence and amount of precipitation, whereas for the analogs both aspects are downscaled simultaneously. More information about the configurations can be found in Tables A1 and 2, or in the companion paper notebook*

4. How the developed package is able to deal with large datasets (10-100-500GB)? Is there any support to larger-than-memory computing (e.g.Python Dask)?

Current on-going work is being done in order to handle larger matrices using the bigmemory package. Also, we are considering future developments in order to be able to run scalable applications in the climate4R Hub (a

cloud-based facility allowing to remotely running climate4R applications). In the meantime, some large tasks can be conveniently sliced using the helper function downscaleChunk. For brevity, we have not included further details on these new developments in this paper. However, there is a related article currently under interactive discussion in this journal in which some of these features are presented. The application of deep learning in downscaling applications is presented and some features to handle large datasets are here presented: https://www.geosci-model-dev-discuss.net/gmd-2019-278/

5. Can the authors say something about the importance of choosing the right domain to compute the EOF? Sometimes the results can be very sensitive to the choice of the domain.

As the referee points out, the domain selection is an important part of model building, being an important decision affecting model performance. In this paper, we show how domain selection can be very easily accomplished with just changing simple parameters (lonLim and latLim) either on the predictor dataset loading (function loadGridData) or by recursively subsetting the already loaded predictor set (using the function subsetGrid). This allows for a flexible configuration of experiments in which different alternative domains can be easily tested. However, in this paper we stick to the domains already well tested in previous studies over the Iberia Peninsula (Gutiérrez et al. 2013) and over Europe, using to this aim the standard experimental settings of the VALUE experiment, to which downscaleR has contributed the methods analysed. Even though domain screening is out of the scope of this article (focused on the presentation of the downscaling tool), we indicate how these type of experiments can be easily undertaken. As an interesting alternative to this time-consuming task, we show that a local predictor approach, based on the use of local predictors close the predictand location can be used without significant changes in the future deltas obtained. To better illustrate this finding, we include further details on the future climate deltas in the new revised section 4.2, and the new Fig. 8 of the revised manuscript shows how the local predictor approach does not significantly alter the deltas obtained.

---

## Author Comment (AC3) · 9 Dec 2019

**Major Issues**

The authors present the R package downscaleR. In principle this is a very useful contribution and worth publishing in GMD. But before publication I ask the authors to address the following major issues, plus a series of minor but still important ones.

In section 4 the authors consider a pan-European setting, and explore whether models using a European predictor domain with additional local predictors perform equally well as the corresponding models with predictors defined on regional domains as used in VALUE. If these models would indeed perform well, this would mean a substantial simplification, e.g., for large-scale ESD applications such as in EURO-CORDEX. I am afraid, however, that the reasoning is not quite stringent. The validation is based on ERA-Interim predictors, which should well represent local predictors

given that local observations have been assimilated. In a GCM context, these local predictors may not fulfill the perfect prog condition, i.e., they may not be bias free. If this were the case,the GCM-based projections could be substantially biased, and the use of local predictors were not permitted. In fact, biases may also affect the climate change signal. I therefore ask the authors to test the PP assumption: first, they should use the historical simulations of their GCM-predictor experiment and check the perfect prog assumption. And second, they should investigate whether the climate change signals simulated by the local implementations differs from those of the VALUE implementations. If the PP assumption was not fulfilled, and/or if the climate change signal was modified, the au-thors should change their conclusions correspondingly. Even in a positive result, the authors should mention that care is required for the reasons given above.

In the new revised manuscript we have addressed this question by evaluating the distributional similarity between GCM and reanalysis predictors. To this aim, we have created maps of the distributional similarity between ERA-Interim and the EC-EARTH historical simulation considering the Kolmogorov-Smirnov (KS) statistic.

The KS statistics are calculated separately for each season (winter and summer), considering the corresponding daily time series for each of the predictor variables and for each grid point. Moreover, in order to isolate distributional dissimilarities due to errors in the first- and second-order moments, we also consider anomalies and standardized anomalies. In the first case, the data are centered by removing the seasonal mean, and in the second case we additionally divide by the seasonal standard deviation. Due to the strong serial correlation present in the daily time series, the test is prone to inflation of type 1 error, that is, rejecting the null hypothesis of equal distributions when it is actually true. To this aim, an effective sample size correction has been applied to the data series to calculate the p-values (Wilks 2006). The methodology followed is similar to the steps followed in Brands *et al.* (2012; 2013).

In perfect-prog statistical downscaling the predictors are rarely used without transformation, and most often data are transformed in such a way that

distributional dissimilarities between reanalysis and GCM can be alleviated. To highlight this fact, we conduct the similarity analysis considering not only the raw time series, but also their corresponding anomalies (i.e., centered data to zero mean) and standardized anomalies (zero mean and unit variance), as introduced in the statistical models used in this paper. The following panels present the overall results. Please note one single summary figure displaying the overall results is presented in the revised manuscript version (Figure 7 of the revised manuscript). For brevity, in the paper the disaggregated results by seasons are omitted, and only two variables, the one that performs best (sea-level pressure) and the one that performs worst (specific humidity at 500 mb) are displayed. The figures displayed below and the code generating them is presented in the companion paper notebook of the revised manuscript. Also, a new measure (ts.ks.pval, https://github.com/SantanderMetGroup/VALUE/blob/devel/R/measure.ks.pval.R) has been introduced in the package VALUE in order to provide the p-values of the KS-test statistic.

In the figures below, the color darkening from pale to deep blue indicate increasing values of the KS-statistic. The significant grid cells (i.e., those for which the distributions of ERA-Interim and EC-EARTH significantly differ), are highlighted with red crosses.

In general terms, the distributions of GCM and reanalysis differ significantly when considering the raw time series, independently of the target season (Figs 1 and 2), thus violating the assumptions of the perfect prog hypothesis regarding the good representativity by the GCM of the reanalysis predictor fields. Centering the data (i.e, zero mean time series) greatly alleviates this problem for most variables, excepting specific humidity at 500 mb (hus@500), and near-surface temperature (tas), persisting some local problems over the British Isles for ta@850 and hus@850 (the latter only in summer, but not in JJA). This is depicted in Figs. 3 (DJF) and 4 (JJA).

Finally, data standardization improves the distributional similarity, attaining an optimal representativity of all the GCM predictors but hus@500 in the summer in southern in the Mediterranean. These results are consistent with

the findings in Brands et al. 2013, pointing to specific humidity in 500 mb as a less reliable predictor, although in the european domain used here problems in the representation of this variable by EC-EARTH are mostly fixed with data standardization.

[Figure]

*Fig. 1. Results of the KS test applied to the time series from the EC-EARTH ESM and ERA-Interim VALUE respectively, considering the original (not transformed) series, for the period 1979-2005 and the DJF season. The grid points showing low p.values (p<0.05) have been marked with a red cross, indicating significant differences in the distribution of both GCM and reanalysis time series.*

**2-sample KS test - Raw series - JJA**

[Figure]

**Fig. 2**. *Same as Fig 1, but for JJA*

**2-sample KS test - Centered anom - DJF**

[Figure]

*Fig. 3*. *Same as Fig 1 (DJF) but using the EC-EARTH and ERA-Interim transformed series, both centered to have zero mean*

[Figure]

**Fig. 4**. *Same as Fig. 3, but for boreal summer JJA.*

[Figure]

*Fig. 5*. *Same as Fig. 3 (DJF), but using standardized anomalies instead of centered anomalies*

**2-sample KS test - Standardized anom - JJA**

*Fig. 6. Same as Fig. 5 (standardized anomalies), but for JJA*

1b. Second, they should investigate whether the climate change signals simulated by the local implementations differs from those of the VALUE implementations. If the PP assumption was not fulfilled, and/or if the climate change signal was modified, the authors should change their conclusions correspondingly. Even in a positive result, the authors should mention that care is required for the reasons given above.

After having verified the perfect-prog assumption regarding the adequate representation of the predictors by the GCM, we have investigated whether the projected climate change deltas are robust to the alternative use of the local predictor approach. Our results indicate that overall, the projected climate change signals for the target indices are not significantly altered.

Figure 7 depicts the relative climate change signals for the local-based (i.e., M1-L and M6-L) and VALUE (i.e., M1 and M6) configurations for the R01 (first row) and SDII (second row) indices. According to the R01 there is consistency among the methods to indicate that a decrease(increase) in the occurrence of precipitation will happen in Southern(Northern) Europe, whereas rainy days will be more intense on average overall in Europe. Slight differences occur when considering the downscaling technique (e.g., M1 and M6) however these differences do not vary as a function of the local predictor configurations taken into account within each downscaling technique. For example, whereas both analogs-based projections present negative relative delta values in the R01 for the Alps, GLM approaches do not predict changes for some of the stations located in the Alps.

[Figure]

*Fig. 7 Relative delta change signals of the R01 and SDII precipitation indices for the future period 2071--2100 (w.r.t. the baseline 1979--2005), obtained by the downscaled projections of the CMIP5 GCM EC-EARTH-r12i1p1, considering the RCP8.5 experiment. The SD methods used are M1-L, M1, M6 and M6-L.*

In conclusion, local-based approaches obtain similar climate change signals for the R01 and SDII indices than the VALUE predictor configurations. There are some differences, but in any case these are smaller between local window/VALUE window than those between GLMs(M1)/Analogs(M6), and therefore the use of the local window does not add additional uncertainty to the climate change signal obtained. Therefore, these results further support the use of local windows centered

on the predictand locations, always subject to the cautionary assessments of the perfect prog hypothesis previously undertaken.

2. I am wondering how downscaleR is placed relative to ESMValTool. This is a widely used tool mainly (but not exclusively) in the GCM community, and it should be possible to combine analyses and results from the different tools. It would be disappointing if the two packages would not be compatible (beyond the exchange of NetCDF files), so a discussion is absolutely necessary, and compatibility very much desired.

ESMValTool is aimed at creating a unified framework for the assessment and evaluation of GCMs. Beyond this primary objective, it exists the possibility of adding further user-tailored layers of functionality by means of the so called "recipes" but, in general, the code is quite complex (using different languages for different modules) and extending the functionality is not straightforward (Moreover, the framework is not fully open source, since there is one private core version). The framework is conceived as a pipeline of data access (via CMOR compliant NetCDF files), post-processing, and evaluation (or recipes). Therefore, the most straightforward way to use ESMValTool is to produce NetCDF files (with downscaled results) and use the standard pipeline (with the standard GCM-oriented validation tools). downscaleR can export the results as NetCDF files, so in principle there is the potential to "integrate" both tools.

downscaleR is envisaged as a fully open specific tool for undertaking statistical downscaling experiments within a single computing environment (R), and seamlessly integrated with other components allowing for the development of end-to-end application, from data retrieval to transformation, visualization, analysis and validation, handling the typical data structures required in most climate applications (that is, regular/irregular gridded datasets and point observations, including additional dimensions such as members and/or initialization times). The whole framework has been branded as "climate4R", and it is since the beginning a completely independent development of the ESMValTool. Of course, this doesn't preclude from an eventual convergence to the ESMValTool workflow, although this idea has not been considered in the development of the different climate4R components.

ESMValTool applies validation measures to files or sets of files based on a convention for file/attribute naming that can be configured via recipes. ESMValTool has a default configuration for CMIP5 and CMIP6 with predefined DRS configurations. Some authors of the manuscript have previous experience in extending ESMValTool with some configurations for CORDEX in the framework of C3S, thus using the tool for the validation of other types of datasets different from GCMs. In principle, and based on this previous experience, it would be possible to  apply the measures defined by ESMValTool to the downscaleR outputs, after export to netcdf using the climate4R tools to this aim (package loadeR.2nc, https://github.com/SantanderMetGroup/loadeR.2nc) using an appropriate recipe to this aim. However, the compatibility of ESMValTool to station data remains as something that requires more time and careful consideration. To our knowledge ESMValTool does not provide support to point data, thus precluding from a straightforward application of downscaling experiments to point stations, as in VALUE.

3. The conclusions are quite weak. I would really appreciate if the authors could discuss what the purpose of the package is, and where it sits in the wide landscape of downscaling and evaluation tools in climate sciences, and what the specific advantages are. This has been touched in the introduction, but here it should be referred back, and some substantial statements should be made.

Following the referee's advice, we have strengthened the conclusions of the manuscript, better highlighting the main features of downscaleR and its unique characteristics within the plethora of tools currently available.

**References**

Brands, S., Gutiérrez, J.M., Herrera, S., Cofiño, A.S., 2012. On the Use of Reanalysis Data for Downscaling. J. Clim. 2517–2526. https://doi.org/10.1175/JCLI-D-11-00251.1

Brands, S., Herrera, S., Fernández, J., Gutiérrez, J.M., 2013. How well do CMIP5 Earth System Models simulate present climate conditions in Europe and Africa?: A performance comparison for the downscaling community. Climate Dynamics 41, 803–817. https://doi.org/10.1007/s00382-013-1742-8

Wilks, D. (2006) Statistical methods in the atmospheric sciences, 2nd ed. Elsevier, Amsterdam

**Minor issues**

In general, some minor grammatical errors (e.g. l 192 "analogs performance") need to be corrected.

l 5: VALUE is a network, not a project. You might also call it an initiative.

Fixed

l 25: "are not suitable" This is not always true. Please replace by "are often not suitable"

Done

l 32: "SD" here you could refer to a recent review or introductory text, e.g., Maraun &Widmann, CUP, 2018.

Done

l 40: "It must be noted" is a zero phrase. Start with "SD techniques are..."

Rephrased

l 45: Here it would be fair to cite Barsugli et al., EOS, 2013.

Thanks for the reference, this has been added

l 55: Here it would be useful to cite the synthesis article, Maraun et al., IJC, 2019,highlighting that this article aims at giving an overall assessment of relative merits and limitations.

Done

l 66: "It is worth mentioning here": Again, a zero phrase. You could rather state "This toolbox complements/adds to other existing tools..."

Done

l 106: somewhere in the introduction you should mention ECMValTool

In this case, we don't see exactly where the ESMValTool fits here. For this reason, we did not include a specific mention to this tool.

l 113: here you should really also refer to Maraun & Widmann, CUP, 2018. It is the most comprehensive discussion of the two approaches in a climate change context.

Done

l 119: no - the term "perfect" refers to the assumption that the predictors are bias free. In particular in weather forecasting, also for MOS the day-to-day correspondence is given. For the MOS discussion you should make clear that the limitation of having homogeneous predictor-predictand relationships applies only in a climate context. This is

also the reason why MOS in climate research is typically just bias correction.  In weather forecasting, you are as free as in PP.

Thanks for the clarification. The text has been modified accordingly

l 130: you may consider presenting the updated assumptions formulated by Maraun &Widmann, CUP, 2019. They are more precise and include the often neglected requirement that the model structure should be applicable.

l 169: you may consider to add a comment that often predictors are proxies for physical processes, which is a main reason for non stationarities in the predictor/predictand relationship, as amply discussed in Maraun & Widmann, CUP, 2019. In this context, you should mention that predictor selection and the training of transfer functions are carried out on short term variability in present climate, whereas the aim is typically to simulate long term changes of short term variability (same reference, and Huth, J. Clim., 2004)

These suggestions have been included

l 194:  it should be pointed out that this is true only for analog methods, which use the same sequence of analogs for different locations.  Otherwise spatial coherence is underestimated. This has been demonstrated by the cited Widmann et al., IJC, 2019.l 196: this statement could be formulated much stronger. I am not aware of any region in the world, where climate change will be so moderate, that the analog method still ap-plies in the far future, when temperature and directly related variables are considered.

These clarifications have been included in the revised text

l 205: somewhere you should mention that the main advantage of GLMs is to simulate(non-normal) variance not explained by the predictors (e.g.,  von Storch,  J. Climate,2000, although, strangely, not all models make use of that).Fig 5: the violin plot needs some explanation. It is not quite clear what the distribution shows. Densities across stations? Is there some kernel smoothing applied? Also: is this an annual analysis? The same holds for the following figures as well.

Violin plots have been explained in more detail in the revised version of the manuscript.

---

## Author Response (AR1)

A step commonly carried out when assessing the 'quality' or 'value' of climate data is the comparison with observed data, normally applying a downscaling step. This paper presents a reproducible R-based workflow in the context of the COST action VALUE. The paper presents a workflow (also shared as R Markdown notebook) which start with data loading to the visualisation of the results. In this workflow the authors compare different downscaling techniques.

I have a few comments here that I think would improve the submitted paper:

1. In the Section 4.1 the authors might add some numbers to Figure 6 (even a separate table) showing average (possibly also std or quantiles) values of RMSE, Correlation and variance ratio. Comparing M1, M6 and their -L version graphically is not easy.

We have included the numbers corresponding to the validation of the pan-european experiment in the new Table 3 (p. 26) of the revised manuscript version.

2. Again in Figure 6 I don't understand the meaning of 'A factor of 0.1 has been applied to RMSE for better comparability of results.', why not leaving the original values?

We decided to apply a scaling factor of 0.1 to the RMSE values in order to make their magnitude comparable to that of the other validation measures, so they can be visually compared in the same plot. We have replace the caption of Fig. 6 (p. 25) indicating that *"[...] The colour bar indicates the mean value of each measure. A factor of 0.1 has been applied to RMSE in order to attain the same order of magnitude in the Y-axis for all the validation measures"*, hoping that it is now more clear.

3. The authors should say something on the computation time needed for the experiments described in the Figure.

We have included a new section in Appendix 1 devoted to a more detailed analysis of computing times. We expect these results to help users to have an approximate idea of the computational effort required for running the different experiments carried-out in the study, also indicating those tasks (mostly predictor configuration and model fitting) that require more time to be accomplished. All the experiments have been undertaken in a single machine and without applying the parallelization option available, so these results can be directly extrapolated to the common situation in which most users run their computations.

[Figure]

*Figure 1. Cross-validation times required for the downscaling models developed in the Iberian experiment. The computational times of the generalized linear models configurations include both the downscaling of the occurrence and amount of precipitation, whereas for the analogs both aspects are downscaled simultaneously. More information about the configurations can be found in Tables A1 and 2 in the Appendix A1 of the revised manuscript.*

4. How the developed package is able to deal with large datasets (10-100-500GB)? Is there any support to larger-than-memory computing (e.g.Python Dask)?

Current on-going work is being done in order to handle larger matrices using the bigmemory package. Also, we are considering future developments in order to be able to run scalable applications in the climate4R Hub (a cloud-based facility allowing to remotely running climate4R applications). In the meantime, some large tasks can be conveniently sliced using the helper function downscaleChunk. For brevity, we have not included further details on these new developments in this paper, given that most statistical downscaling applications commonly used in impact studies are undertaken at local/regional

scales and therefore do not handle huge datasets. However, there is a related article currently under interactive discussion in this journal in which some of these features are presented. The application of deep learning in downscaling applications is presented and some features to handle large datasets are here presented: https://www.geosci-model-dev-discuss.net/gmd-2019-278/. We have included citation to this article in the new revised version of the manuscript, L193.

5. Can the authors say something about the importance of choosing the right domain to compute the EOF? Sometimes the results can be very sensitive to the choice of the domain.

As the referee points out, the domain selection is an important part of model building, being an important decision affecting model performance. In this paper, we show how domain selection can be very easily accomplished with just changing simple parameters (lonLim and latLim) either on the predictor dataset loading (function loadGridData) or by recursively subsetting the already loaded predictor set (using the function subsetGrid). This allows for a flexible configuration of experiments in which different alternative domains can be easily tested. However, in this paper we stick to the domains already well tested in previous studies over the Iberia Peninsula (Gutiérrez *et al.* 2013) and over Europe, using to this aim the experimental protocols of the VALUE experiment, to which downscaleR has contributed with the methods here analysed. Even though domain screening is out of the scope of this article (focused on the presentation of the downscaling tool), we indicate how these type of experiments can be easily undertaken. As an interesting alternative to this time-consuming task, we show that a local predictor approach, based on the use of local predictors close the predictand location, can be used without significant changes in the future deltas obtained. This is one of the key results of this study, and so has been highlighted in the conclusions. To better illustrate this finding, we include further details on the future climate deltas in the new revised section 4.2, and the new Fig. 8 of the revised manuscript shows how the local predictor approach does not significantly alter the deltas obtained. We explicitly indicate that this approach has the additional

advantage of avoiding the task of optimal sub-domain selection for predictor configuration (including EOF calculation).

**Interactive comment on "Statistical downscaling with the downscaleR package: Contribution to the VALUE intercomparison experiment"**

https://doi.org/10.5194/gmd-2019-224-RC2

Response to reviewer #2

We thank the referee for her/his time and the insightful feedback provided. In this document we include a point-by-point response to the comments received. The new revised version of the manuscript includes a number of modifications following the referee's advice, in which we have invested considerable effort and interest. We hope that the referee will deem the revised manuscript version of sufficient quality for publication. In this response, the referee's comments are indicated in black, and the author responses in blue fonts.

**Major Issues**

The authors present the R package downscaleR. In principle this is a very useful contribution and worth publishing in GMD. But before publication I ask the authors to address the following major issues, plus a series of minor but still important ones.

In section 4 the authors consider a pan-European setting, and explore whether models using a European predictor domain with additional local predictors perform equally well as the corresponding models with predictors defined on regional domains as used in VALUE. If these models would indeed perform well, this would mean a substantial simplification, e.g., for large-scale ESD applications such as in EURO-CORDEX. I am afraid, however, that the reasoning is not quite stringent. The validation is based on ERA-Interim predictors, which should well represent local predictors

given that local observations have been assimilated. In a GCM context, these local predictors may not fulfill the perfect prog condition, i.e., they may not be bias free. If this were the case,the GCM-based projections could be substantially biased, and the use of local predictors were not permitted. In fact, biases may also affect the climate change signal. I therefore ask the authors to test the PP assumption: first, they should use the historical simulations of their GCM-predictor experiment and check the perfect prog assumption. And second, they should investigate whether the climate change signals simulated by the local implementations differs from those of the VALUE implementations. If the PP assumption was not fulfilled, and/or if the climate change signal was modified, the au-thors should change their conclusions correspondingly. Even in a positive result, the authors should mention that care is required for the reasons given above.

In the new revised manuscript we have addressed this question by evaluating the distributional similarity between GCM and reanalysis predictors. To this aim, we have created maps of the distributional similarity between ERA-Interim and the EC-EARTH historical simulation considering the Kolmogorov-Smirnov (KS) statistic. We have also reinforced Section 2 of the manuscript (p. 4-6) in order to provide a more thorough description of the perfect-prog assumptions, and we have included a new Section 4.2.1 (p.25-27) in which the ability of the GCM to represent the predictors is carefully analysed.

The results from this analysis are summarized in Fig. 7 of the new revised manuscript, although more details for all predictor variables and a seasonal analysis is included in the companion paper notebook (Section Code and Data Availability, updated). For brevity, in the paper the disaggregated results by seasons are omitted, and only two variables, the one that performs best (sea-level pressure) and the one that performs worst (specific humidity at 500 mb) are displayed. The companion paper notebook has been updated, and further details on how to perform this analysis have been included. This new analysis has entailed an update to the core package VALUE to include a new measure implementing the corrected p-value of the KS score.

1b. Second, they should investigate whether the climate change signals simulated by the local implementations differs from those of the VALUE implementations. If the PP assumption was not fulfilled, and/or if the climate change signal was modified, the authors should change their conclusions correspondingly. Even in a positive result, the authors should mention that care is required for the reasons given above.

After having verified the perfect-prog assumption regarding the adequate representation of the predictors by the GCM, we have investigated whether the projected climate change deltas are robust to the alternative use of the local predictor approach. Our results indicate that overall, the projected climate change signals for the target indices are not significantly altered, and that the SD method (GLM vs. analogs) adds much more uncertainty to the projected deltas that the predictor configuration approach. These results are consistent with previous findings by San-Martón et al. 2016, as indicated in the new revised version of the manuscript (Section 4.2.2). In conclusion, our results further support the use of local windows centered on the predictand locations, always subject to the cautionary assessments of the perfect prog hypothesis previously undertaken, as the referee points-out.

2. I am wondering how downscaleR is placed relative to ESMValTool. This is a widely used tool mainly (but not exclusively) in the GCM community, and it should be possible to combine analyses and results from the different tools. It would be disappointing if the two packages would not be compatible (beyond the exchange of NetCDF files), so a discussion is absolutely necessary, and compatibility very much desired.

ESMValTool is aimed at creating a unified framework for the assessment and evaluation of GCMs. Beyond this primary objective, it exists the possibility of adding further user-tailored layers of functionality by means of the so called "recipes" but, in general, the code is quite complex (using different languages for different modules) and extending the functionality is not straightforward (Moreover, the framework is not fully open source, since there is one private core version). The framework is conceived as a pipeline of data access (via CMOR compliant NetCDF files), post-processing, and evaluation (or recipes). Therefore, the most

straightforward way to use ESMValTool is to produce NetCDF files (with downscaled results) and use the standard pipeline (with the standard GCM-oriented validation tools). downscaleR can export the results as NetCDF files, so in principle there is the potential to "integrate" both tools.

downscaleR is envisaged as a fully open specific tool for undertaking statistical downscaling experiments within a single computing environment (R), and seamlessly integrated with other components allowing for the development of end-to-end application, from data retrieval to transformation, visualization, analysis and validation, handling the typical data structures required in most climate applications (that is, regular/irregular gridded datasets and point observations, including additional dimensions such as members and/or initialization times). The whole framework has been branded as "climate4R", and it is since the beginning a completely independent development of the ESMValTool. Of course, this doesn't preclude from an eventual convergence to the ESMValTool workflow, although this idea has not been considered in the development of the different climate4R components.

ESMValTool applies validation measures to files or sets of files based on a convention for file/attribute naming that can be configured via recipes. ESMValTool has a default configuration for CMIP5 and CMIP6 with predefined DRS configurations. Some authors of the manuscript have previous experience in extending ESMValTool with some configurations for CORDEX in the framework of C3S, thus using the tool for the validation of other types of datasets different from GCMs. In principle, and based on this previous experience, it would be possible to apply the measures defined by ESMValTool to the downscaleR outputs, after export to netcdf using the climate4R tools to this aim (package loadeR.2nc, https://github.com/SantanderMetGroup/loadeR.2nc) using an appropriate recipe to this aim. However, the compatibility of ESMValTool to station data remains as something that requires more time and careful consideration. To our knowledge ESMValTool does not provide support to point data, thus precluding from a straightforward application of downscaling experiments to point stations, as in VALUE.

3. The conclusions are quite weak. I would really appreciate if the authors could discuss what the purpose of the package is, and where it sits in the wide landscape of downscaling and evaluation tools in climate sciences, and what the specific advantages are. This has been touched in the introduction, but here it should be referred back, and some substantial statements should be made.

Following the referee's advice, we have strengthened the conclusions of the manuscript, better highlighting the main features of downscaleR and its unique characteristics within the plethora of tools currently available.

**Minor issues**

In general, some minor grammatical errors (e.g. l 192 "analogs performance") need to be corrected.
l 5: VALUE is a network, not a project. You might also call it an initiative.
Fixed. The imprecise term VALUE "Project" has been replaced by initiative/framework throughout the manuscript.
l 25: "are not suitable" This is not always true. Please replace by "are often not suitable"
Done (L25)
l 32: "SD" here you could refer to a recent review or introductory text, e.g., Maraun &Widmann, CUP, 2018.
Done (L32)
l 40: "It must be noted" is a zero phrase. Start with "SD techniques are..."
Rephrased
l 45: Here it would be fair to cite Barsugli et al., EOS, 2013.
Thanks for the reference, this has been added (L45)
l 55: Here it would be useful to cite the synthesis article, Maraun et al., IJC, 2019,highlighting that this article aims at giving an overall assessment of relative merits and limitations.
Done (L54)
l 66: "It is worth mentioning here": Again, a zero phrase. You could rather state "This toolbox complements/adds to other existing tools..."
Done

l 106: somewhere in the introduction you should mention ECMValTool

For the reasons explained above (major comment 2), we don't see exactly where the ESMValTool fits here and its relationship with downscaleR. For this reason, we did not include a specific mention to this tool.

l 113: here you should really also refer to Maraun & Widmann, CUP, 2018. It is the most comprehensive discussion of the two approaches in a climate change context.

Done (L115)

l 119: no - the term "perfect" refers to the assumption that the predictors are bias free. In particular in weather forecasting, also for MOS the day-to-day correspondence is given. For the MOS discussion you should make clear that the limitation of having homogeneous predictor-predictand relationships applies only in a climate context. This is also the reason why MOS in climate research is typically just bias correction. In weather forecasting, you are as free as in PP.

130: you may consider presenting the updated assumptions formulated by Maraun &Widmann, CUP, 2019. They are more precise and include the often neglected requirement that the model structure should be applicable.

l 169: you may consider to add a comment that often predictors are proxies for physical processes, which is a main reason for non stationarities in the predictor/predictand relationship, as amply discussed in Maraun & Widmann, CUP, 2019. In this context, you should mention that predictor selection and the training of transfer functions are carried out on short term variability in present climate, whereas the aim is typically to simulate long term changes of short term variability (same reference, and Huth, J. Clim., 2004)

Thanks for the clarifications and valuable comments above. The whole section has been rewritten to include all the referee's suggestions and clarifications regarding the characteristics and application of MOS and PP techniques. (L125-140).

l 194: it should be pointed out that this is true only for analog methods, which use the same sequence of analogs for different locations. Otherwise spatial coherence is underestimated. This has been demonstrated by the cited Widmann et al., IJC, 2019.

Corrected. (L208-210)

l 196: this statement could be formulated much stronger. I am not aware of any region in the world, where climate change will be so moderate, that the

analog method still ap-plies in the far future, when temperature and directly related variables are considered.

The statement has been reformulated accordingly (L214)

l 205: somewhere you should mention that the main advantage of GLMs is to simulate(non-normal) variance not explained by the predictors (e.g., von Storch, J. Climate,2000, although, strangely, not all models make use of that).Fig 5: the violin plot needs some explanation. It is not quite clear what the distribution shows. Densities across stations? Is there some kernel smoothing applied? Also: is this an annual analysis? The same holds for the following figures as well.

Violin plots have been explained in more detail in the revised version of the manuscript (L560-572). Please not that the inputs for each validation measure are indicated in Table 1. In the caption of each violin.plot Figure (Figs.5 and 6) we indicate the sample size (number of stations) for each violin.

[revised manuscript text omitted]